# Repression of hypoxia-inducible factor-1 contributes to increased mitochondrial reactive oxygen species production in diabetes

Xiaowei Zheng[1†], Sampath Narayanan[1†], Cheng Xu[1†], Sofie Eliasson Angelstig[1], Jacob Grünler[1], Allan Zhao[1], Alessandro Di Toro[2], Luciano Bernardi[3], Massimiliano Mazzone[4], Peter Carmeliet[5], Marianna Del Sole[1], Giancarlo Solaini[6], Elisabete A Forsberg[1], Ao Zhang[1§], Kerstin Brismar[1], Tomas A Schiffer[7], Neda Rajamand Ekberg[1,8,9], Ileana Ruxandra Botusan[1,8,9], Fredrik Palm[7‡], Sergiu-Bogdan Catrina[1,8,9*‡]

[1]Department of Molecular Medicine and Surgery, Karolinska Institutet, Stockholm, Sweden; [2]Centre for Inherited Cardiovascular Diseases, IRCCS Foundation University Hospital Policlinico San Matteo, Pavia, Italy; [3]Folkälsan Research Center, Folkälsan Institute of Genetics, University of Helsinki, Helsinki, Finland; [4]Laboratory of Tumor Inflammation and Angiogenesis, Center for Cancer Biology, Vlaams Instituut voor Biotechnologie (VIB); Laboratory of Tumor Inflammation and Angiogenesis, Center for Cancer Biology, Department of Oncology, Katholieke Universiteit (KU) Leuven, Leuven, Belgium; [5]Laboratory of Angiogenesis and Vascular Metabolism, Department of Oncology, Katholieke Universiteit (KU) Leuven; Laboratory of Angiogenesis and Vascular Metabolism, Vesalius Research Center, Vlaams Instituut voor Biotechnologie (VIB), Leuven, Belgium; [6]Dipartimento di Biochimica, Università di Bologna, Bologna, Italy; [7]Department of Medical Cell Biology, Uppsala University, Uppsala, Sweden; [8]Department of Endocrinology and Diabetes, Karolinska University Hospital, Stockholm, Sweden; [9]Center for Diabetes, Academic Specialist Centrum, Stockholm, Sweden

*For correspondence:
sergiu-bogdan.catrina@ki.se

†These authors contributed equally to this work
‡These authors also contributed equally to this work

Present address: §Department of Nephrology, GuangDong Second Traditional Chinese Medicine Hospital, GuangZhou, China

Competing interest: The authors declare that no competing interests exist.

## Abstract

**Background:** Excessive production of mitochondrial reactive oxygen species (ROS) is a central mechanism for the development of diabetes complications. Recently, hypoxia has been identified to play an additional pathogenic role in diabetes. In this study, we hypothesized that ROS overproduction was secondary to the impaired responses to hypoxia due to the inhibition of hypoxia-inducible factor-1 (HIF-1) by hyperglycemia.

**Methods:** The ROS levels were analyzed in the blood of healthy subjects and individuals with type 1 diabetes after exposure to hypoxia. The relation between HIF-1, glucose levels, ROS production and its functional consequences were analyzed in renal mIMCD-3 cells and in kidneys of mouse models of diabetes.

**Results:** Exposure to hypoxia increased circulating ROS in subjects with diabetes, but not in subjects without diabetes. High glucose concentrations repressed HIF-1 both in hypoxic cells and in kidneys of animals with diabetes, through a HIF prolyl-hydroxylase (PHD)-dependent mechanism. The impaired HIF-1 signaling contributed to excess production of mitochondrial ROS through increased mitochondrial respiration that was mediated by Pyruvate dehydrogenase kinase 1 (PDK1). The

restoration of HIF-1 function attenuated ROS overproduction despite persistent hyperglycemia, and conferred protection against apoptosis and renal injury in diabetes.

**Conclusions:** We conclude that the repression of HIF-1 plays a central role in mitochondrial ROS overproduction in diabetes and is a potential therapeutic target for diabetic complications. These findings are timely since the first PHD inhibitor that can activate HIF-1 has been newly approved for clinical use.

**Funding:** This work was supported by grants from the Swedish Research Council, Stockholm County Research Council, Stockholm Regional Research Foundation, Bert von Kantzows Foundation, Swedish Society of Medicine, Kung Gustaf V:s och Drottning Victorias Frimurarestifelse, Karolinska Institute's Research Foundations, Strategic Research Programme in Diabetes, and Erling-Persson Family Foundation for S-B.C.; grants from the Swedish Research Council and Swedish Heart and Lung Foundation for T.A.S.; and ERC consolidator grant for M.M.

## Editor's evaluation

The paper is novel, informative, and with interesting translational implications. This paper will be of interest to scientists interested in diabetes and its complications, as well as the wider field of hypoxia biology. It provides evidence to understand why diabetes causes damage to multiple tissues when oxygen supply becomes limited.

## Introduction

Excessive production of mitochondrial ROS is a key contributor to oxidative stress, which is a major cause of diabetic complications (*Charlton et al., 2020*; *Giacco and Brownlee, 2010*). In diabetes, excessive production of ROS in mitochondria is caused by an increased proton gradient across the mitochondrial membrane. This occurs secondary to elevated electron transport chain flux, mainly at complex I and complex III (*Nishikawa et al., 2000*).

Hypoxia also plays an important role in the development of diabetic complications and is present in both patients with diabetes (*Bernardi et al., 2011*) and in animal models with diabetes, in all tissues in which complications occur (*Catrina, 2014*; *Catrina and Zheng, 2021*). Hypoxia-inducible factor-1 (HIF-1) is a transcription factor central in the cellular response to low oxygen tension (*Prabhakar and Semenza, 2015*). HIF-1 is a heterodimeric transcription factor composed of two subunits, HIF-1α and HIF-1β, both of which are ubiquitously expressed in mammalian cells. Regulation of HIF-1 function is critically dependent on the degradation of the HIF-1α subunit in normoxia. The molecular basis of its degradation is oxygen-dependent hydroxylation of at least one of the two proline residues by specific $Fe^{2+}$-, and 2-oxoglutarate-dependent HIF prolyl hydroxylases (PHD 1–3), among which, PHD2 (encoded by *Egln1* gene) has the main role. Hydroxylated HIF-1α binds to the von Hippel–Lindau (VHL) tumour suppressor protein, which acts as an E3 ubiquitin ligase and targets HIF-1α for proteasomal degradation. Under hypoxic conditions, HIF-1α is stabilized against degradation, translocates to the nucleus, binds to hypoxic responsive elements (HRE) and activates transcription of a series of genes involved in different processes (i.e. angiogenesis, cell proliferation, survival, and cell metabolism). These processes enable the cell to adapt to reduced oxygen availability (*Schödel and Ratcliffe, 2019*).

HIF-1, as the key mediator of adaptation to low oxygen tension, contributes to a balance in redox homeostasis by supressing the excessive mitochondrial production of ROS under chronic hypoxia, thereby minimizing potentially deleterious effects (*Semenza, 2017*). Since HIF-1 stability and function is complexly repressed in diabetes (*Catrina and Zheng, 2021*), we hypothesized that its repression might contribute to increased ROS. We therefore investigated the impact of glucose levels on ROS production during hypoxia in cells, animal models of diabetes and patients with diabetes, and whether the excessive mitochondrial ROS production in diabetes could be normalized by restoring HIF-1 function.

We found that repressed HIF-1 function secondary to hyperglycemia contributes to an overproduction of mitochondrial ROS with direct pathogenic effects. Consequently, pharmacological or genetic interventions to prevent repression of HIF-1 function normalize mitochondrial production of ROS in

diabetes and inhibit the development of nephropathy, in which hypoxia plays an important pathogenic role (*Haase, 2017*).

## Materials and methods

Key Resources Table is in Appendix 1 - key resources table.

### Clinical study

Thirteen non-smoking patients with type 1 diabetes (28.9 ± 7.2 years old; 53.8% male and 46.2% female; HbA1c: 74.4 ± 11.8 mmol/mol (9.0% ± 1.1 %); BMI: 24.3 ± 4.0 kg/m$^2$) and 11 healthy, age-matched controls (30.5 ± 8.5 years old; 54.5% male and 45.5% female; HbA1c: 35.6 ± 2.6 mmol/mol (5.4% ± 0.2 %); BMI: 24.3 ± 4.0 kg/m$^2$) were exposed to intermittent hypoxia for 1 hr, consisting of five hypoxic episodes (13% $O_2$, 6 min) that alternated with normoxic episodes (20.9% $O_2$, 6 min) (*Figure 1—figure supplement 1*). The subject breathed through a disposable mouthpiece which was connected via an antibacterial filter (Carefusion, Yorba Linda, CA,USA) to a T-tube carrying one-way respiratory valves (Tyco Healthcare, Hamshire, UK). A nose clip assured that respiration occured through the mouth. The air supplied to the subject came from a tube connected to a stopcock (Hans Rudolph, Shawnee, KS, USA) connected to a 60 L Douglas bag (Hans Rudolph) which was continuously filled with hypoxic gas. By turning the stopcock, the air supplied to the subject could be switched from hypoxic to ambient (normoxic) air. The Douglas bag was positioned behind the subject's bed, so that he/she could not notice when the hypoxic gas or normoxic air was supplied. Blood samples were taken before and immediately after hypoxia exposure. Patients had been diagnosed with diabetes for 10–20 years, showed no signs of peripheral neuropathy and had intact peripheral sensibility when checked with monofilament and vibration tests. The study was approved by the Regional Ethical Review Board in Stockholm, Sweden, and carried out in accordance with the principles of the Declaration of Helsinki. The sample size has been decided according to the experience from previous studies (*Duennwald et al., 2013*). All participants in the study provided informed consent.

### EPR spectroscopy

ROS levels in the blood were measured using Electron Paramagnetic Resonance (EPR) Spectroscopy (*Dikalov et al., 2018*). Blood samples were mixed with spin probe 1-hydroxy-3-carboxy-pyrrolidine (CPH, 200 µM) in EPR-grade Krebs HEPES buffer supplemented with 25 mM Deferoxamine (DFX) and 5 mM diethyldithiocarbamate (DETC), and incubated at 37°C for 30 min before being frozen in liquid nitrogen. EPR measurements were carried out using a table-top EPR spectrometer (Noxygen Science Transfer & Diagnostics GmbH, Elzach, Germany). The spectrometer settings were as follows: microwave frequency, 9.752 GHz; modulation frequency, 86 kHz; modulation amplitude, 8.29 G; sweep width, 100.00 G; microwave power, 1.02 mW; number of scans, 15. All data were converted to absolute concentration levels of CP radical (mmol $O_2^-$/min/µg) using the standard curve method. All chemicals and reagents for EPR Spectroscopy were obtained from Noxygen Science Transfer & Diagnostics GmbH.

### Cell culture

Mouse Inner Medullary Collecting Duct-3 (mIMCD-3) cells (ATCC CRL-2123; ATCC, USA) were cultured in Dulbecco's modified Eagle's medium (DMEM; 5.5 mM glucose) supplemented with 10% heat-inactivated FBS and 100 IU/ml penicillin and streptomycin (Thermo Fisher Scientific). The cells were maintained in a humidified atmosphere with 5% $CO_2$ at 37°C in a cell culture incubator, and were tested negatively for microplasma using MycoAlert PLUS mycoplasma detection kit (LONZA). Cells were cultured under normoxic (21% $O_2$) or hypoxic (1% $O_2$) conditions in Hypoxia Workstation INVIVO2 (Ruskinn).

### Nuclear extraction

To detect HIF-1α, mIMCD-3 cells were cultured in medium containing 5.5 or 30 mM glucose for 24 hours in the absence or presence of DMOG (200 µM), and were exposed to normoxia or hypoxia for 6 hours prior to harvest. The cells were collected and incubated on ice for 10 min in hypotonic buffer containing 10 mM KCl, 1.5 mM $MgCl_2$, 0.2 mM PMSF, 0.5 mM dithiothreitol, and protease

inhibitor mix (Complete-Mini; Roche Biochemicals). After the cells were swollen, nuclei were released using a Dounce homogenizer Type B. The nuclei were pelleted and resuspended in a buffer containing 20 mM Tris (pH 7.4), 25% glycerol, 1.5 mM MgCl$_2$, 0.2 mM EDTA, and 0.02 M KCl. Soluble nuclear proteins were released from the nuclei by gentle, drop-wise addition of a buffer containing 20 mM Tris (pH 7.4), 25% glycerol, 1.5 mM MgCl$_2$, 0.2 mM EDTA, and 0.6 M KCl, followed by 30 min of incubation in ice. The nuclear extracts were then centrifuged and dialyzed in dialysis buffer containing 20 mM Tris (pH 7.4), 20% glycerol, 100 mM KCl, 0.2 mM EDTA, 0.2 mM PMSF, 0.5 mM dithiothreitol, and protease inhibitor mix.

## Plasmid construction and transfection

Plasmids encoding an HRE-driven luciferase reporter, Renilla luciferase, GFP, and GFP-HIF-1α were described previously (*Zheng et al., 2006*). Plasmid pCMV3-FLAG-PDK1 encoding FLAG-tagged human PDK1 was obtained from Sino Biological Inc (Catalog number: HG12312-NF). pCMV3-GFP-FLAG-PDK1 encoding GFP-fused FLAG-tagged PDK1 was generated by subcloning a Hind III-GFP-Hind III fragment from pCMV2-FLAG-GFP into Hind III - digested pCMV3-FLAG-PDK1. Plasmid transfection was performed using Lipofectamine reagent (Thermo Fisher Scientific) according to the manufacturer's protocol.

## HRE-driven luciferase reporter assay

HIF-1 activity was determined by an HRE-driven luciferase reporter assay. mIMCD-3 cells were transfected with plasmids encoding HRE-driven firefly luciferase and Renilla luciferase using Lipofectamine reagent. Cells were then cultured in media containing normal (5.5 mM) or high (30 mM) glucose concentrations, and were exposed to normoxia or hypoxia for 40 hours. The cells were harvested, and luciferase activity was measured using the Dual Luciferase Assay System (Promega) on the GloMax Luminometer (Promega) according to the manufacturer's instructions. HRE-driven firefly luciferase activity was normalized to Renilla luciferase activity and expressed as relative luciferase activity.

## Cellular apoptosis analysis

mIMCD3 cells were cultured in media containing normal (5.5 mM) or high (30 mM) glucose concentrations and were exposed to normoxia or hypoxia for 24 hours before analysis. Apoptosis was analyzed using Annexin V-FITC / 7-AAD kit (Beckman Coulter) according to the manufacturer's protocol. Briefly, the cells were incubated with Annexcin V-FITC and 7-AAD for 15 min in the dark, and then analyzed within 30 min using flow cytometry on a Cyan ADP analyser (Beckman Coulter). The gating scheme is shown in *Figure 2—figure supplement 1*. Results were expressed as percentage of Annexin V – positive and 7-AAD – negative apoptotic cells.

## Determination of caspase 3/7 activity

mIMCD3 cells were seeded in 96-well plate with 1500 cells/well in duplicates, and were cultured in media containing normal (5.5 mM) or high (30 mM) glucose concentrations and were exposed to normoxia or hypoxia for 24 hr before analysis. Caspase 3/7 activity was evaluated using Caspase-Glo 3/7 assay kit (Promega) on the GloMax Luminometer (Promega) according to the manufacturer's instructions. The caspase 3/7 activity was finally normalized to the DNA concentration in each well using Quant-iT dsDNA High-Sensitivity Assay Kit (Thermo Fisher Scientific) measured using GloMax Discover Microplate Reader (Promega).

## RNA interference

siRNA for mouse VHL (Flexitube Gene Solution GS22346) was obtained from Qiagen. AllStars negative control siRNA, obtained from Ambion, was used as a control. siRNA was transfected using Lipofectamine RNAiMAX Transfection Reagent (Thermo Fisher Scientific), according to the manufacturer's protocol. Twenty-four hours after transfection, cells were exposed to 5.5 or 30 mM glucose in normoxia or hypoxia for 24 hr before being harvested.

## Detection of mitochondrial ROS levels using flow cytometry

After 24 hr' exposure to 5.5 or 30 mM glucose levels in normoxia or hypoxia, mIMCD-3 cells were stained with MitoSOX Red Mitochondrial Superoxide Indicator (Thermo Fisher Scientific). A working

**Table 1.** Characteristics of *Lepr*^db/db^ (db/db) and control mice prior to experiments.

| Groups | WT-Control | Db/db-Control | Db/db-DMOG |
|---|---|---|---|
| Body weight (g) | 27.44 ± 0.41 | 48.66 ± 1.00 | 50.04 ± 1.08 |
| Blood glucose (mM) | 7.16 ± 0.39 | 21.27 ± 1.21 | 20.59 ± 1.50 |
| Age (weeks) | 16 ± 0 | 17.25 ± 0.48 | 17.00 ± 0.45 |
| n | 14 | 16 | 16 |

Data are presented as mean ± SEM. Source data are shown in Table 1—source data 1.

The online version of this article includes the following source data for table 1:

**Source data 1.** Characteristics of *Lepr*^db/db^ and control mice prior to experiments.

concentration of 5 μM was used, and cells were incubated at 37°C for 10 min protected from light. After washing off excess dye, cells were trypsinized and suspended in Krebs HEPES buffer and analysed using flow cytometry on a Cyan ADP analyser (Beckman Coulter). Analysis was performed using FlowJo software, and the gating scheme is shown in *Figure 3—figure supplements 1 and 2*. Mitochondrial ROS levels were expressed as percentage of MitoSOX Red fluorescence intensity.

## Fluorescent immunocytochemistry and confocal microscopy

mIMCD3 cells were seeded on coverslips and transfected with siRNA or plasmids as desired and were exposed to hypoxia and high glucose levels for 24 hr. The cells were then fixed in 4% Formaldehyde (Sigma) at room temperature (RT) for 15 min. After three washes with Phosphate-buffered Saline (PBS, Sigma), the cells were premeabilized in PBS containing 0.1% Triton-X100 at RT for 10 min. After three washes with PBS, the cells were blocked with PBS containing 5% Bovine Serum Albumin (BSA, Sigma), and incubated with Rabbit polyclonal anti-HIF-1α antibody (GeneTex, Cat No. GTX127309) 1:200 diluted in PBS containing 1% BSA and 0.1% Tween-20 (Sigma) at 4°C over night. After three washes with PBS containing 0.1% Tween-20 (PBS-T) for 5 min each, the cells were incubated with a fluorochrome-conjugated secondary antibody, Goat anti-Rabbit Alexa 594 (ThermoFisher Scientific, A-11037, 1:500 diluted) in PBS containing 1% BSA and 0.1% Tween-20 at RT for 1 hr. The cells were then washed with PBS-T twice and with PBS twice before the cover slips were mounted on the slides using ProLong Gold Antifade Mountant with DAPI (ThermoFisher Scientific). The fluorescent images were captured using a Leica SP8 confocal microscope (Leica Microsystems).

## Animals

Diabetic male BKS-*Lepr*^db/db^/JOrlRj (*Lepr*^db/db^) mice and healthy controls were from Janvier Labs. Characteristics of the mice prior to experiments are summarized in *Table 1*. *Lepr*^db/db^ mice with HbA1c levels > 55 mmol/mol or blood glucose >15 mM when HbA1c levels were between 45 and 55 mmol/mol were included in the analysis. Mice were allocated into groups according to their age, HbA1c or blood glucose levels. Mice were injected intraperitoneally (*i.p.*) with DMOG (320 mg/kg body weight) 4 days and 1 day before sacrifice for the analysis of mitochondrial function. For other analyses, *Lepr*^db/db^ mice were injected (*i.p.*) with DMOG (50 mg/kg body weight) every second day for 1 month

**Table 2.** Characteristics of *Egln1*^+/-^ and WT mice prior to experiments.

| Groups | WT-Control | *Egln1*^+/-^-Control | WT-diabetic | *Egln1*^+/-^-diabetic |
|---|---|---|---|---|
| Start body weight (g) | 28.02 ± 1.04 | 27.11 ± 0.79 | 28.27 ± 0.71 | 28.64 ± 0.87 |
| Age (weeks) | 22.46 ± 0.85 | 23.91 ± 0.72 | 22.67 ± 0.87 | 24.57 ± 0.59 |
| n | 24 | 23 | 24 | 21 |

Data are presented as mean ± SEM. Source data are shown in Table 2—source data 1.

The online version of this article includes the following source data for table 2:

**Source data 1.** Characteristics of *Egln1*^+/-^ and WT mice prior to experiments.

**Table 3.** Blood glucose of *Egln1*[+/-] and WT mice before and after STZ injection.

| Groups | WT-diabetic | *Egln1*[+/-]-diabetic |
|---|---|---|
| Blood glucose (mM) before STZ | 5.11 ± 0.24 | 4.18 ± 0.20 |
| Blood glucose (mM) after STZ | 19.39 ± 1.04 | 17.7 ± 0.81 |
| n | 24 | 21 |

Data are presented as mean ± SEM. Source data are shown in Table 3—source data 1.

The online version of this article includes the following source data for table 3:

**Source data 1.** Blood glucose of *Egln1*[+/-] and WT mice before and after STZ injection.

before sacrifice. *Egln1*[+/-] mice and their wild-type (WT) littermates were generated as previously described (*Mazzone et al., 2009*). Characteristics of the mice prior to experiments are shown in *Table 2*. Diabetes was induced in male *Egln1*[+/-] and WT mice with streptozotocin (STZ) *i.p.* injections. STZ was administered at 50 mg/kg body weight daily for five consecutive days, and mice were diabetic for at least 6 weeks before sacrifice. Blood glucose before and after STZ injection are shown in *Table 3*. Mice were exposed to a 12 hr light/dark cycle at 22°C, and were given standard laboratory food and water ad libitum. The sample size was calculated to achieve 30% difference in albuminuria between DMOG or vehicle – treated *Lepr*[db/db] mice or between diabetic *Egln1*[+/-] and WT mice and was adjusted for each parameter according to preliminary results. The experimental animal procedure was approved by the North Stockholm Ethical Committee for the Care and Use of Laboratory Animals.

## Fluorescent immunohistochemistry

Formalin-fixed, paraffin-embedded kidney tissues were deparaffinized and rehydrated, and antigen retrieval was performed in citrate buffer using a pressure cooker. After washing the slides with PBS-T three times for 3 min each, sections were demarcated with a hydrophobic pen. Sections were blocked with goat serum in PBS for 30 min at RT and then incubated overnight at 4 °C with primary antibodies (HIF-1α antibody, GeneTex, GTX127309, 1:100 diluted; KIM-1 antibody, Novus Biologicals, NBP1-76701, 1:50 diluted). Sections were then washed with PBS-T four times for 5 min each. Sections were incubated for 1 hr at RT in the dark with a fluorochrome-conjugated secondary antibody, Goat anti-Rabbit Alexa fluor 488 or 594 (ThermoFisher Scientific, A-11008 or A-11037, 1:500 diluted). Sections were then washed with PBS-T four times for 5 min each and treated with 0.1% Sudan Black-B solution (Sigma) for 10 min to quench autofluorescence. Sections were counterstained with DAPI for 3 min, and were mounted and stored at 4 °C. Fluorescent images were acquired using a Leica TCS SP5 and SP8 confocal microscope (Leica Microsystems). Image analysis was blinded and performed using Image-Pro Premier v9.2 (Media Cybernetics) software.

To detect hypoxia in mouse kidneys, pimonidazole solution (Hypoxyprobe–1 Omni Kit, Hypoxyprobe, Inc) was *i.p.* administered to mice at a dosage of 60 mg/kg body weight 90 min prior to tissue harvest. Pimonidazole adducts were detected on kidney sections using a 1:100 diluted RED PE dye-conjugated mouse monoclonal anti-pimonidazole antibody (clone 4.3.11.3) according to the Hypoxyprobe RED PE Kit protocol.

To detect HIF-1α, the above method was modified to incorporate the Tyramide Superboost kit (Thermo Fisher Scientific). Briefly, after antigen retrieval, the sections were blocked with 3% $H_2O_2$ to quench endogenous peroxidase activity before blocking with goat serum (both ingredients provided in the kit). Following the PBS-T washes after primary HIF-1α antibody incubation, the sections were incubated with an HRP-conjugated rabbit antibody for 1 hr at RT. Sections were washed rigorously and incubated with tyramide reagent for 10 min at RT in the dark. The reaction was stopped by incubating with the stop solution for 5 min, and samples were washed with PBS-T three times for 3 mins each. The sections were subsequently treated with 0.1% Sudan Black-B solution, counterstained and mounted as mentioned above.

## Evaluation of ROS levels in kidney

ROS levels in kidney tissues were indirectly assessed by evaluating non-enizmatic lipid peroxidation by measuring 4-Hydroxynonenal (4-HNE) protein adduct levels using the OxiSelect HNE Adduct Competitive ELISA kit (STA838, Cell Biolabs) according to the manufacturer's instruction.

## Kidney mitochondrial function

Mitochondria were isolated from mouse kidneys, and mitochondrial function was determined using high-resolution respirometry (Oxygraph 2 k, Oroboros) as previously described (*Schiffer et al., 2018*). The analysis was blinded. Briefly, respirometry was performed in respiration medium containing EGTA (0.5 mM), $MgCl_2$ (3 mM), K-lactobionate (60 mM), taurine (20 mM), $KH_2PO_4$ (10 mM), HEPES (20 mM), sucrose (110 mM), and fatty-acid-free BSA (1 g/L). Pyruvate (5 mM) and malate (2 mM) were added to measure state two respiration, followed by the addition of ADP (2.5 mM) to measure complex I-mediated maximal respiratory capacity (state three respiration). Complex I + II-mediated maximal oxidative phosphorylation was evaluated after adding succinate (10 mM). LEAK respiration was measured in the presence of pyruvate (5 mM), malate (2 mM), and oligomycin (2.5 μM). Respiration was normalized to mitochondrial protein content, determined spectrophotometrically using the DC Protein Assay kit (Bio-Rad).

## RNA purification and quantitative RT-PCR

Total RNA was extracted from kidney using miRNeasy Mini kit (Qiagen). cDNA was produced using High-Capacity cDNA Reverse Transcription Kit (Thermo Fisher Scientific). Quantitative RT-PCR was performed on a 7300 or 7900 Real-Time PCR System (Applied Biosystems) using SYBR Green Master Mix (ThermoFisher Scientific). The average gene expression of β-actin (*ACTB*) and Hydroxymethyl-bilane synthase (*HMBS*) was used as control. Primer sequences are listed in Key Resources Table in Appendix 1 - key resources table.

## Protein extraction and western blotting

Kidney biopsies were homogenized in a buffer containing 50 mM Tris-HCl (pH 7.4), 180 mM NaCl, 0.2% NP-40, 20% glycerol, 0.5 mM phenylmethylsulfonyl fluoride, 5 mM β-mercaptoethanol, and a protease inhibitor mix (Complete-Mini; Roche Biochemicals). Cell lysate was obtained by centrifugation for 30 min at 20,000 g and 4°C. Protein concentrations were determined using the Bradford Protein Assay (Bio-Rad) according to the manufacturer's protocol. Nuclear extracts and tissue lysate were separated by SDS-PAGE and blotted onto nitrocellulose membranes. Blocking was performed in TBS buffer (50 mM Tris pH 7.4 and 150 mM NaCl) containing 5% nonfat milk, followed by incubation with anti-HIF-1α (1:500, NB100-479; Novus Biologicals), anti-Histone H3 (1:5000, ab1791; Abcam), anti-KIM-1 (1:500, NBP1-76701; Novus Biologicals) or anti-α-tubulin (1:1000, MAB11106; Abnova) antibodies in TBS buffer containing 1% nonfat milk. After several washes, the membranes were incubated with IRDye 800 goat anti-rabbit or IRDye 680 goat anti-mouse secondary antibodies (LI-COR). The membranes were then scanned with Odyssey Clx Imaging System (LI-COR). Quantification of western blots were performed using ImageJ (version 1.53).

## TUNEL staining

Apoptosis in kidneys was detected using the In Situ Cell Death Detection Kit (Sigma Aldrich/Roche). Briefly, formalin-fixed paraffin-embedded sections were deparaffinized, rehydrated and blocked with 3% $H_2O_2$ to quench endogenous peroxidase activity. The sections were permeabilized with 0.1% Triton X-100, 0.1% sodium citrate solution and blocked with 3% BSA in PBS; the sections were then incubated with the TUNEL mixture for 1 hr at 37°C. Sections were thoroughly rinsed in PBS, treated with 0.1% Sudan Black-B solution to quench autofluorescence and counterstained with DAPI. Sections were mounted and stored at 4°C. Images were obtained using a Leica TCS SP8 confocal microscope (Leica Microsystems). The images were analyzed using Image-Pro Premier v9.2 software (Media Cybernetics). TUNEL-positive nuclei were counted and expressed as a percentage of the total number of nuclei.

## Albuminuria

Urine was collected from mouse bladders after sacrifice and snap frozen in liquid nitrogen. Urine albumin and creatinine concentrations were evaluated in thawed urine samples using a DCA Vantage

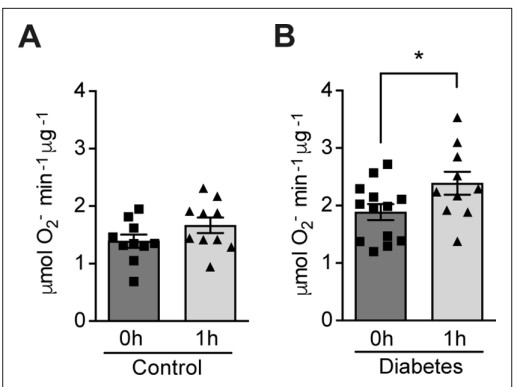

**Figure 1.** Hypoxia increases circulating ROS in patients with diabetes but not in control subjects without diabetes. Healthy controls (**A**) and subjects with type 1 diabetes (**B**) were exposed to intermittent hypoxia for 1 hr. Peripheral blood was taken before (0h) and after (1h) hypoxia exposure. ROS levels were analyzed using Electron Paramagnetic Resonance (EPR) Spectroscopy with CPH spin probes (n = 10–13). Data are represented as mean ± SEM. *, p < 0.05 analysed using unpaired two-sided Student's t-test. This figure has one figure supplement. Source data are shown in *Figure 1—source data 1*.

The online version of this article includes the following source data and figure supplement(s) for figure 1:

**Source data 1.** ROS levels in blood from patients with diabetes and control subjects.

**Figure supplement 1.** Schematic demonstration of hypoxia exposure protocol in the clinical study.

Analyzer (Siemens Healthcare GmbH) with the corresponding test cartridges DCA Microalbumin/Creatinine ACR urine test (01443699, Siemens Healthcare GmbH).

## Statistical analysis

All data used for statistical analysis are independent biological replicates. Technical replicates were applied during luciferase reporter, ELISA, caspase 3/7 activity, DNA and protein concentration, and QPCR analysis; and the average of the results from technical replicates is regarded as one biological data. Statistical analysis was performed using GraphPad Prism software. Outliers identified using Grubbs' test were excluded from analysis. The differences between two groups were analysed using unpaired two-sided Student's t-test. Multiple comparisons of three or more groups were performed using one-way or two-way ANOVA followed by Bonferroni's post hoc test or Holm–Sidak's test, or Brown-Forsythe and Welch ANOVA tests followed by Dunnett T3 multiple comparison test for sample set with unequal standard deviations. p < 0.05 was considered statistically significant. Data are presented as mean ± standard error of the mean (SEM).

## Results

### Hypoxia increases circulating ROS in patients with diabetes but not in control subjects without diabetes

The effect of hypoxia on ROS production was evaluated in patients with poorly controlled type 1 diabetes (28.9 ± 7.2 years old; HbA1c: 74.4 ± 11.8 mmol/mol) and matched control subjects without diabetes (30.5 ± 8.5 years old; HbA1c: 35.5 ± 2.6 mmol/mol). Participants were exposed to mild and intermittent hypoxia (13% $O_2$) for 1 hr (*Figure 1—figure supplement 1*), which is known to elicit a clinical response (*Duennwald et al., 2013*). As shown in *Figure 1*, ROS levels in peripheral blood were increased by hypoxia in patients with diabetes. However, hypoxia did not change the ROS levels in normoglycemic control subjects.

### High glucose concentrations inhibit HIF-1 signaling through PHD-dependent mechanism and induce apoptosis in hypoxia

Since hypoxia induces ROS in diabetes, and HIF-1 is the central regulator of cellular responses to hypoxia (*Prabhakar and Semenza, 2015*), we hypothesized that the dysregulated HIF-1 signaling contributes to the ROS overproduction in diabetes. We tested this hypothesis using mouse inner medulla collecting tubular cells (mIMCD-3), given the important pathogenic role of hypoxia in diabetic kidney disease (*Palm, 2006*). As shown in *Figure 2A*, the nuclear expression of HIF-1α increased after exposure to hypoxia; however, this effect was attenuated under hyperglycemic conditions. Moreover, HRE-driven luciferase reporter assay showed less HIF activity in hypoxia under hyperglycemic conditions compared with normoglycemic conditions (*Figure 2B*). High glucose concentrations also increased apoptosis of mIMCD3 cells during hypoxia (*Figure 2C*). Interestingly, when the cells were exposed to dimethyloxalylglycine (DMOG), a competitive inhibitor of PHD, both HIF-1α expression (*Figure 2A*) and HIF-1 function (*Figure 2D*) were increased and apoptosis was inhibited (*Figure 2E*)

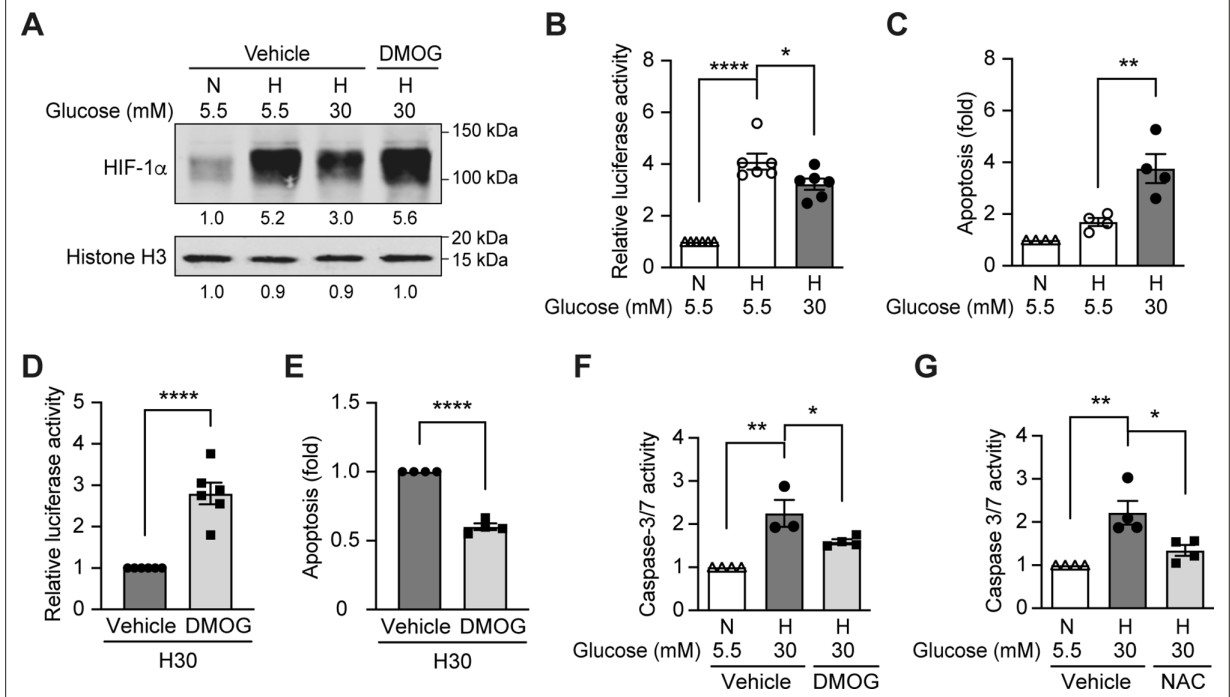

**Figure 2.** High glucose levels inhibit HIF-1 signaling and induce apoptosis, which can be rescued by PHD inhibitor DMOG. (**A**) mIMCD-3 cells were cultured in normal (5.5 mM) or high (30 mM) glucose media in the presence of DMOG or vehicle for 24 hr, and were exposed to hypoxia (**H**) or normoxia (**N**) for 6 hr before harvest. The nuclear expression of HIF-1α and Histone H3 was measured using western blotting. (**B–F**) mIMCD-3 cells were exposed to 5.5 or 30 mM glucose levels in normoxia (**N**) or hypoxia (**H**) in the presence or absence of DMOG or vehicle for 24 hr. The relative HRE-driven luciferase activity (**B and D**, n = 6), apoptosis (**C and E**, n = 4), and the caspase 3/7 activity (**F**, n = 3–4) were assessed. (**G**) Caspase 3/7 activity was evaluated in mIMCD-3 cells that were pre-treated with 1 mM NAC or vehicle for 1 hr before exposure to 5.5 or 30 mM glucose levels in normoxia (**N**) or hypoxia (**H**) for 24 hr (n = 4). The data under control conditions were considered as 1.0. Data are shown as mean ± SEM. *, p < 0.05; **, p < 0.01; ***, p < 0.001; ****, p < 0.0001 using one-way ANOVA followed by Bonferroni's post hoc test (**B–C, F–G**), and unpaired two-sided Student t-test (**D–E**). This figure has one figure supplement. Source data are shown in *Figure 2—source data 1*.

The online version of this article includes the following source data and figure supplement(s) for figure 2:

**Source data 1.** HRE-driven luciferase activity, apoptosis and caspase 3/7 activity in mIMCD3 cells.

**Figure supplement 1.** Flow cytometry gating strategy for the evaluation of cellular apoptosis.

under hypoxic and hyperglycemic conditions. These results indicate that high glucose levels inhibit HIF-1 and induce apoptosis in hypoxic mIMCD3 cells through a PHD-dependent mechanism. Moreover, high glucose levels in hypoxia also enhanced caspase-3/7 activity in mIMCD3 cells, which could be inhibited by DMOG treatment (*Figure 2F*), suggesting that the apoptosis induced by high glucose levels and hypoxia is dependent on caspase-3 and –7.

We next assessed the role of ROS in mediating apoptosis induced by hyperglycemia under hypoxic conditions, by pretreatment of the mIMCD3 cells with the thiol reducing agent N-acetylcysteine (NAC). As shown in *Figure 2G*, pretreatment with NAC significantly inhibited the increase of caspase-3/7 activity in mIMCD3 cells exposed to high glucose levels and hypoxia, indicating the role of ROS in the induction of apoptosis in these conditions.

## Repression of HIF-1 by high glucose concentrations contributes to increased mitochondrial ROS production in hypoxia

We further investigated the influence of HIF-1 on mitochondrial ROS production in diabetes. Mitochondrial ROS levels were increased in cells exposed to high glucose levels and hypoxia (*Figure 3A*), which corresponded to impaired HIF-1 activity. Interestingly, HIF-1 activation by DMOG diminished the mitochondrial ROS overproduction induced by high glucose levels in hypoxia (*Figure 3A*), indicating HIF-1 repression as an important mechanism for increased mitochondrial ROS production in diabetes. This

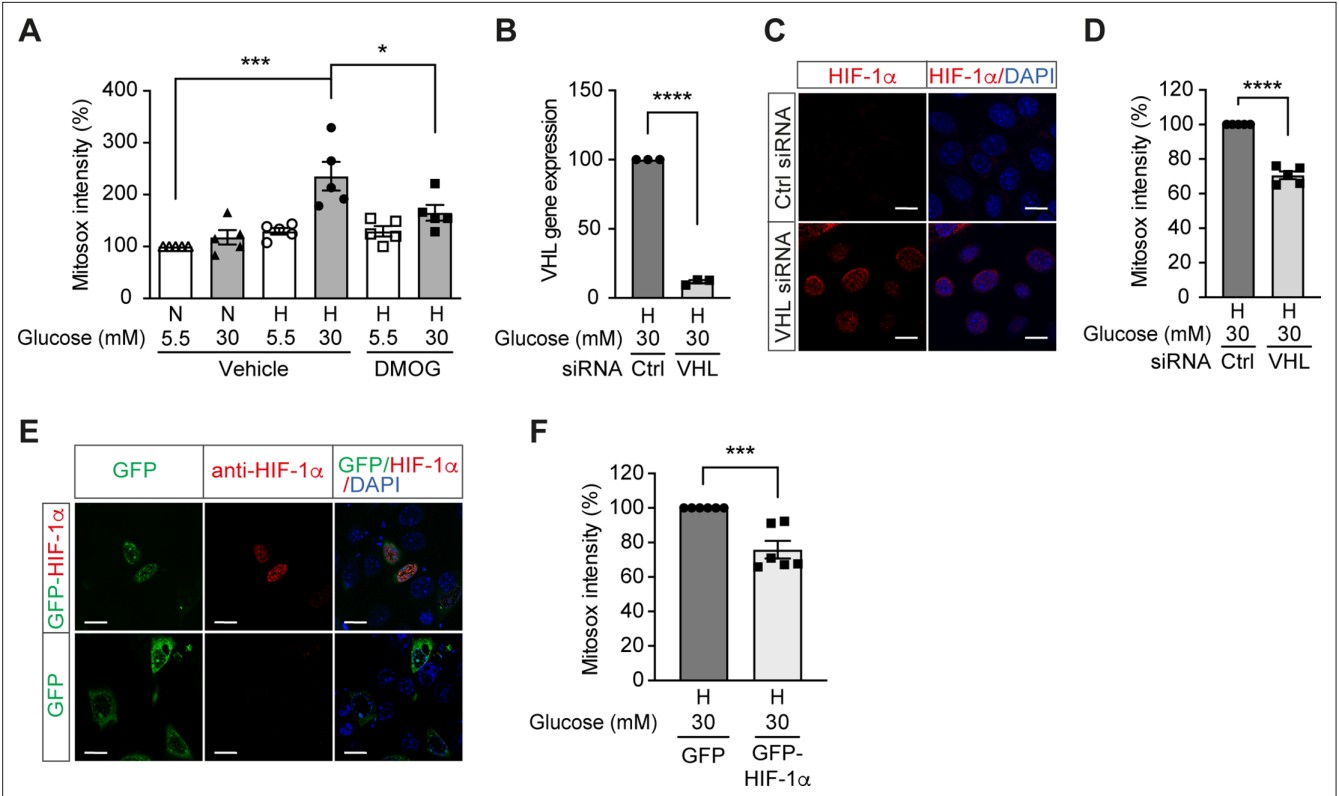

**Figure 3.** High glucose levels induce mitochondrial ROS overproduction in hypoxia, which can be rescued by promoting HIF-1 function. (**A**) Mitochondrial ROS levels were measured as mitosox intensity in mIMCD-3 cells cultured in normal (5.5 mM) or high (30 mM) glucose media in normoxia (N) or hypoxia (H) for 24 hr in the presence of DMOG or vehicle (n = 5). (**B–D**) mIMCD-3 cells were transfected with von Hippel–Lindau tumour suppressor (VHL) or control (Ctrl) siRNA, and exposed to hypoxia (**H**) and 30 mM glucose for 24 hr. VHL gene expression (**B**, n = 3), endogenous HIF-1α expression (red) and DAPI staining (blue) (**C**) and mitochondrial ROS levels (**D**, n = 5) were assessed using quantitative RT-PCR, fluorescent immunocytochemistry and flow cytometry, respectively. (**E and F**) mIMCD-3 cells were transfected with plasmids encoding GFP or GFP-HIF-1α,and exposed to hypoxia and 30 mM glucose for 24 hr. (**E**) Expression of GFP and GFP-HIF-1α (green) were detected using confocal microscopy. The nuclear HIF-1α expression was confirmed by immucytochemistry using anti-HIF-1α antibody (red). Nuclei were stained blue with DAPI. (**F**) Mitochondrial ROS levels are shown (n = 6). The mitosox intensity of cells cultured under control conditions were considered as 100%. Data are shown as mean ± SEM. *, p < 0.05; ***, p < 0.001; ****, p < 0.0001 using one-way ANOVA followed by Bonferroni's post hoc test (**A**), and unpaired two-sided Student t-test (**B, D and F**). This figure has two figure supplements. Source data are shown in *Figure 3—source data 1*. Scale bar: 50 μm.

The online version of this article includes the following source data and figure supplement(s) for figure 3:

**Source data 1.** Mitosox intensity and VHL gene expression in mIMCD3 cells.

**Figure supplement 1.** Flow cytometry gating strategy for the evaluation of mitosox intensity.

**Figure supplement 2.** Flow cytometry gating strategy for the evaluation of mitosox intensity in mIMCD3 cells transfected with plasmids encoding GFP or GFP-fused protein.

was further confirmed by similar results that were observed when HIF-1 activity was maintained during hyperglycemia in hypoxia by genetic approaches, that is silencing VHL that mediates HIF-1α degradation (*Figure 3B–D*) or overexpressing HIF-1α (*Figure 3E–F*). Silencing VHL gene (*Figure 3B*) in mIMCD3 cells exposed to hypoxia and high glucose levels was followed by an increase of nuclear HIF-1α expression (*Figure 3C*) and lead to decreased mitochondrial ROS (*Figure 3D*). Mitochondrial ROS was also decreased in mIMCD3 cells expressing GFP-HIF-1α compared to cells expressing GFP under hypoxic and hyperglycemic conditions (*Figure 3E–F*). Taken together, these results suggest that mitochondrial ROS overproduction in cells exposed to a combination of hypoxia and hyperglycemia is dependent on the impairment of HIF-1 function and can be attenuated when HIF-1 activity is maintained.

## HIF-1 repression is responsible for excess ROS production in diabetic kidney

To investigate the relevance of HIF-1 modulation on ROS levels in diabetes, we further focused our investigation on the kidney, where low oxygen levels play an important pathogenic role (*Palm et al., 2003*). ROS levels were higher in the kidney from mouse models of both type 2 diabetes (*Lepr^db/db* mice) and streptozotocin (STZ)-induced type 1 diabetes, as evaluated by 4-Hydroxynonenal (HNE) levels (*Figure 4B and D*). At the same time, HIF-1 signaling was repressed, as shown by insufficient activation of HIF-1α (*Figure 4A and C*), despite a profound hypoxic environment indicated by pimonidazole staining (*Figure 4—figure supplement 1*). This reverse correlation between ROS and HIF-1 activity further supports the hypothesis that the repression of HIF-1 signaling contributes to the ROS overproduction in diabetes.

We therefore sought to assess the influence of promoting HIF-1 function during hyperglycemia on ROS production in these animals. To this end, we inhibited PHD activity, either through pharmacological inhibition, by treatment of the *Lepr^db/db* mice with DMOG or through genetic modification by employing *Egln1^+/-* mice in the STZ-induced model of diabetes. Both methods were able to increase HIF-1α levels (*Figure 4A and C*) and HIF-1 activity, as assessed by HIF-1 target gene expression, despite persistence of hyperglycemia (*Figure 4—figure supplements 2 and 3*). Importantly, HIF-1 activation in the kidney was followed by a decrease in renal ROS levels in both *Lepr^db/db* mice and STZ-induced diabetic mice (*Figure 4B and D*).

Investigation of mitochondrial respiration in the kidneys of both animal models revealed an increase of the complex I- and complex I + II-mediated state three respiration and mitochondrial leak (*Figure 4E and F*). Promoting HIF-1 activity in the kidney of diabetic animals, by either DMOG treatment or by haplodeficiency of *Egln1*, was followed by normalization of perturbed mitochondrial respiration (*Figure 4E and F*). Pyruvate dehydrogenase kinase 1 (PDK1), a direct HIF-1 target gene that inhibits the flux of pyruvate through tricarboxylic acid cycle (TCA) and subsequent mitochondrial respiration (*Kim et al., 2006*), was down-regulated in diabetic kidneys and could be rescued by HIF-1 activation (*Figure 4G and H*). These results indicate an important role of PDK1 in mediating the effects of HIF-1 on the regulation of ROS production in diabetic kidney. In order to verify this mechanism, we transfected plasmids encoding GFP or GFP-fused PDK1 (GFP-PDK1) in mIMCD3 cells exposed to high glucose levels in hypoxia (*Figure 4I*), and assessed the mitochondrial ROS levels using flow cytometry analysis of mitosox intensity in GFP- or GFP-PDK1-positive cells. As shown in *Figure 4J*, PDK1 overexpression diminished the mitochondrial ROS overproduction in cells exposed to high glucose levels in hypoxia, suggesting that the increased mitochondrial ROS is at least partially mediated by the inhibition of HIF-1 target gene PDK1 in these conditions.

## Promoting HIF-1 function reduces renal injury and ameliorates renal dysfunction in mouse models of diabetes

Promoting HIF-1 function in diabetic animals, with its secondary suppression of ROS production, exerted protective effects on kidney function. In both *Lepr^db/db* mice (*Figure 5A–C , and G*) and mice with STZ-induced diabetes (*Figure 5D–F , and H*), promoting HIF-1 function prevented typical diabetic kidney lesions, as measured by reduced Kidney Injury Marker-1 (KIM-1) expression (*Figure 5A–B , and D–E*) and TUNEL staining-assessed apoptosis (*Figure 5C and F*). This resulted in improved renal function, as demonstrated by decreased albuminuria in both mouse models of diabetes (*Figure 5G–H*).

## Discussion

Excessive mitochondrial ROS production is a central pathogenic contributor to the development of diabetic complications. In addition, excessive ROS stimulate several other deleterious biochemical pathways such as activation of protein kinase C, formation of advanced glycation end-products, polyol pathway flux and overactivity of the hexosamine pathway (*Nishikawa et al., 2000*). Here, we show that in diabetic models, overproduction of ROS from mitochondria is not due to increased electron transport chain flux secondary to hyperglycemia alone. Impairment of HIF-1 signaling is also a critical mechanism, since promoting HIF-1 activity in diabetic models in vitro and in vivo attenuated ROS production, despite the persistence of hyperglycemia, which prevents the development of oxidative stress-induced kidney injury.

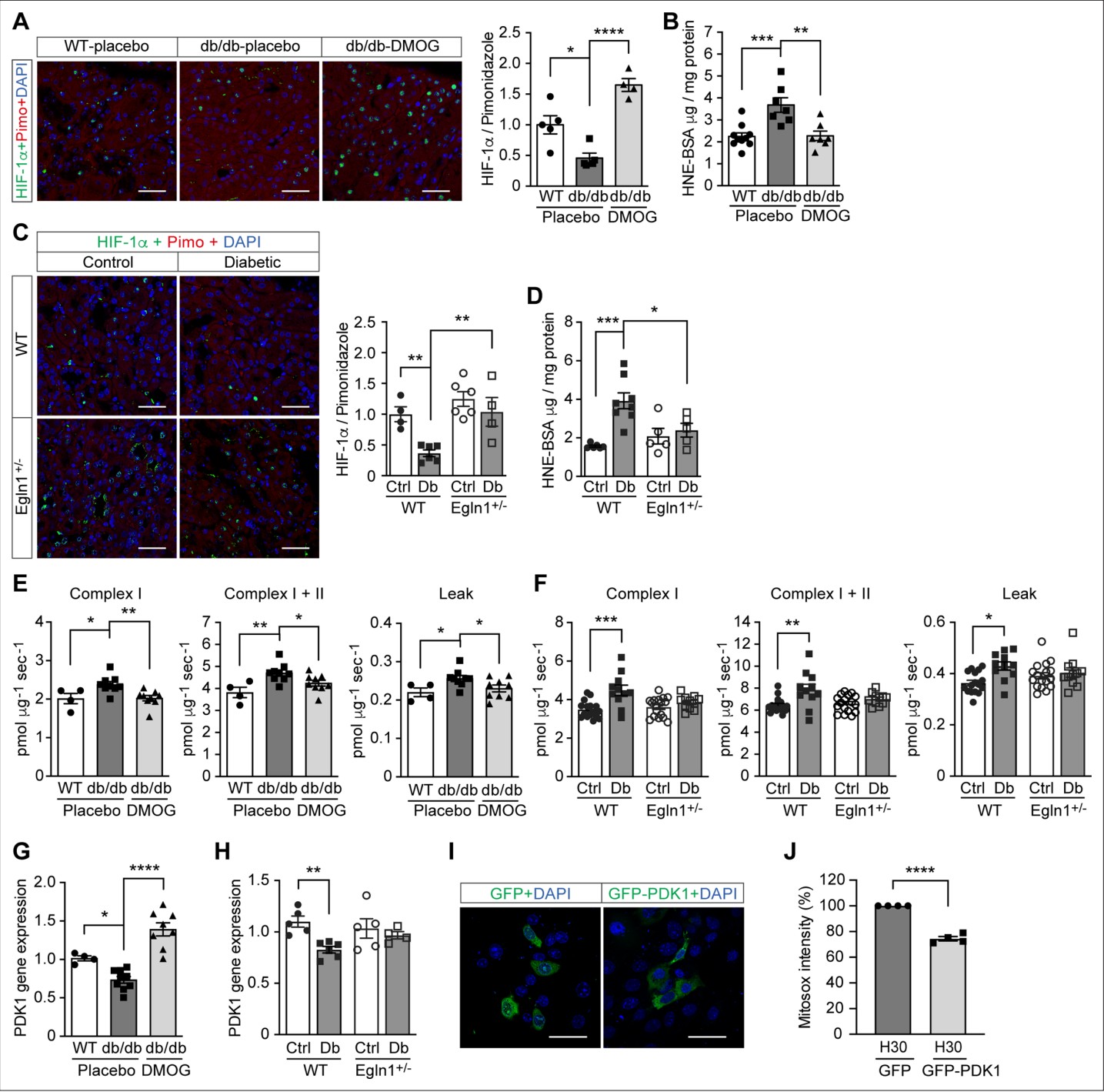

**Figure 4.** Promoting HIF-1 function attenuates renal ROS excess and mitochondrial respiration in mouse models of diabetes. Kidneys were harvested from wild-type (WT) and *Lepr*^db/db^ diabetic mice (db/db) that were treated with placebo (vehicle) or DMOG (**A–B, E, G**), and from non-diabetic control (Ctrl) or diabetic (Db) wild-type (WT) and *Egln1*^+/-^ mice (**C–D, F, H**). (**A and C**) HIF-1α (green), pimonidazole (red, hypoxia marker) and DAPI (blue, nuclear staining) signals were detected by fluorescent immunohistochemistry, and relative HIF-1α expression levels were quantified (**A**, n = 4–5; **C**, n = 4–6). Scale bar: 100 μm. (**B and D**) Renal ROS levels were detected using the OxiSelect HNE adduct competitive ELISA kit (**B**, n = 7–10; **D**, n = 5–8). (**E and F**) Mitochondrial respiratory function was evaluated using high resolution respirometry (**E**, n = 4–9; **F**, n = 11–17). (**G and H**) PDK1 gene expression in kidneys (**G**, n = 4–9; **H**, n = 4–6). (**I and J**) mIMCD-3 cells were transfected with plasmids encoding GFP or GFP-PDK1,and exposed to hypoxia and 30 mM glucose (H30) for 24 hr. (**I**) Expression of GFP and GFP-HIF-1α (green) and nuclear DAPI staining (blue) were detected using confocal microscopy. Scale bar: 50 μm. (**J**) Mitochondrial ROS levels are shown (n = 4). Data are shown as mean ± SEM. *, p < 0.05; **, p < 0.01; ***, p < 0.001; ****, p < 0.0001 using one-way ANOVA (**A, B, E, G**) and two-way ANOVA (**C, D, F, H**) followed by multi-comparison post hoc tests, and unpaired two-sided Student

*Figure 4 continued on next page*

*Figure 4 continued*

t-test (**J**). This figure has three figure supplements. Source data are shown in *Figure 4—source data 1*.

The online version of this article includes the following source data and figure supplement(s) for figure 4:

**Source data 1.** HIF-1α, ROS, and mitochondrial respiration levels in mouse kidneys and PDK1 gene expression and Mitosox intensity in mIMCD3 cells.

**Figure supplement 1.** Kidney in diabetes is more hypoxic.
**Figure supplement 1—source data 1.** Quantification of Pimonidazole immunofluorescent signal in mouse kidneys.

**Figure supplement 2.** DMOG increases HIF-1 target gene expression in *Lepr*[db/db] mice without affecting blood glucose levels.
**Figure supplement 2—source data 1.** Blood glucose and HIF-1 target gene expression levels in *Lepr*[db/db] mice.

**Figure supplement 3.** *Egln1* haplodeficiency increases HIF-1 target gene expression in diabetic mice without affecting blood glucose levels.
**Figure supplement 3—source data 1.** HbA1c and gene expression levels in *Egln1*[+/-] and WT mice.

In subjects with diabetes, ROS levels increase after exposure to hypoxia in opposition to control subjects. Acute hypoxia can unmask the impaired HIF-1 signaling that presents in patients with diabetes. However, other ROS sources responsive to acute hypoxia, either from mitochondria (*Hernansanz-Agustín et al., 2020*; *Waypa et al., 2013*) or from elsewhere (*Weissmann et al., 2000*) cannot be excluded. The concentration of oxygen in tissues ranges from 1% to 10%, which continuously activates HIF-1 signaling machinery (*Carreau et al., 2011*). Therefore, the small decrease in oxygen tension present in patients with diabetes (*Bernardi et al., 2017*), combined with an impaired HIF-1 activation (*Catrina et al., 2004*), may contribute to increased ROS levels in tissues associated with diabetic complications.

Indeed, the direct relationship between hyperglycemia-dependent repression of HIF-1 signaling and excess ROS in hypoxia was demonstrated experimentally both in vitro and in vivo in this study. We found that HIF-1 signaling was inhibited by hyperglycemia in tubular cells during hypoxia and in kidneys from mouse models of diabetes, through a PHD-dependent mechanism, which is in accordance with previous observations (*Bento and Pereira, 2011*; *Catrina, 2014*). This was followed by increased ROS production in mitochondria, when assessed by a specific mitochondrial probe (*Wang et al., 2010*), which was not evident under hyperglycemic conditions when HIF-1 function was promoted with different approaches. The relationship between HIF-1 and ROS is bidirectional, with most evidence showing that mitochondrial ROS has a stabilizing effect on HIF-1α (*Brunelle et al., 2005*; *Chandel et al., 1998*; *Guzy et al., 2005*), although the exact mechanisms are still unclear. Several mechanisms in the repressive effects of HIF-1 signaling on mitochondrial ROS production have also been identified (*Fukuda et al., 2007*; *Kim et al., 2006*). Our results indicate that the role of HIF-1 on ROS is due to the decreased respiration rate of the mitochondria, since both pharmacological and genetic induction of HIF-1 prevents increased respiration. This effect is at least partially mediated by HIF-1 target gene PDK1, that has been previously shown to inhibit pyruvate dehydrogenase (PDH) activity, leading to reduced flux of lactate through TCA cycle and electron transport chain (*Kim et al., 2006*).

Along with the cellular systems that mitigate the effect of ROS (e.g. Nrf2) (*Jiang et al., 2010*), the increased mitochondrial leak noted in diabetes is a pathway aimed at diminishing ROS production by decreasing mitochondrial membrane potential (*Echtay et al., 2002*; *Miwa and Brand, 2003*). However, to produce enough ATP, this is followed by increased flux through the electron transport chain. Although HIF-1 activity normally suppresses electron transport chain, this regulation is diminished in diabetes, resulting in an increased oxygen consumption rate and aggravation of cellular hypoxia that contributes to tissue injury. This was confirmed by increased pimonidazole staining in diabetic kidney in this study, as previously observed (*Rosenberger et al., 2008*). Thus, our results provide evidence for repressed HIF-1 in diabetes as a critical mechanism underlying the vicious cycle between oxidative stress and hypoxia, which is suggested to contribute to kidney injury (*Honda et al., 2019*).

Indeed, pharmacological or genetic interventions to sustain HIF-1 signaling in diabetes normalized ROS production and had direct consequences on kidney function, despite persistent hyperglycemia. Albuminuria, a typical marker of diabetic nephropathy, was prevented in animal models of either type 1 or type 2 diabetes when HIF-1 signaling was maintained. This is in accordance with the previous

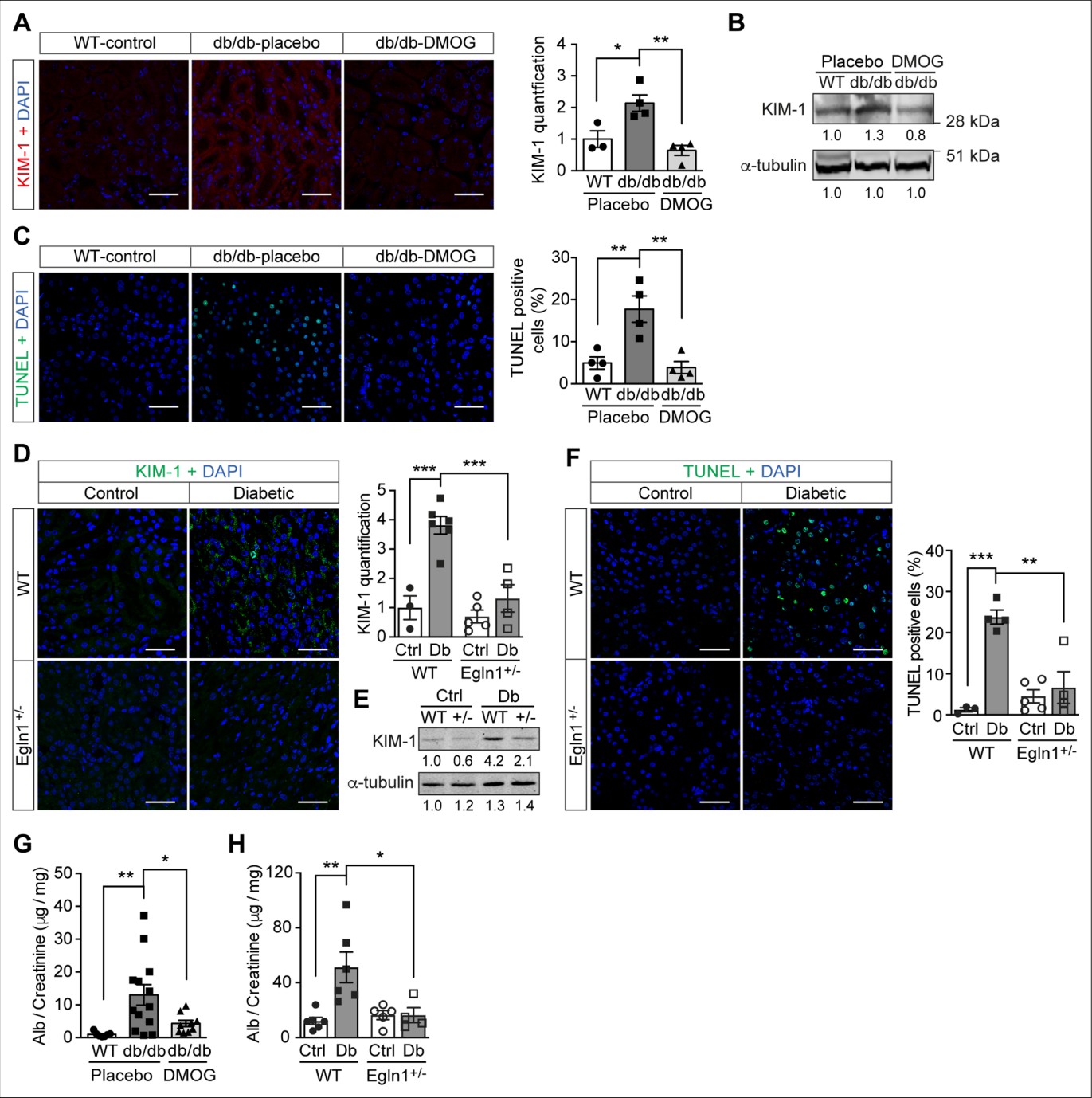

**Figure 5.** Promoting HIF-1 function reduces renal injury and ameliorates renal dysfunction in mouse models of diabetes. Kidneys were harvested from wild-type (WT) and *Lepr^db/db* diabetic mice (db/db) that were treated with placebo (vehicle) or DMOG (**A–C, G**), and from non-diabetic control (Ctrl) or diabetic (Db) wild-type (WT) and *Egln1^+/-* (+/-) mice (**D–F, H**). (**A and D**) Representative images of KIM-1 (red or green) and DAPI (blue) in kidney that were analysed using fluorescent immunohistochemistry. Quantifications of KIM-1 fluoresent signal are shown in corresponding histogram (**A**, n = 3–4; **D**, n = 3–6). (**B and E**) Representative images of KIM-1 and α-tubulin analyzed by western blotting. (**C and F**) Apoptotic cells were detected using TUNEL staining, and the percentage of TUNEL-positive cells were quantified (**C**, n = 4; **F**: n = 3–5). (**G and H**) Albuminuria is presented as the ratio of albumin (Alb) to creatinine in mouse urine (**G**, n = 7–13; **H**, n = 4–6). Data are shown as mean ± SEM. *, p < 0.05; **, p < 0.01; ***, p < 0.001 analysed using one-way ANOVA (**A, C**), Brown-Forsythe and Welch ANOVA (**G**) and two-way ANOVA (**D, F, H**) followed by multi-comparison test. Source data are shown in *Figure 5—source data 1*. Scale bar: 100 µm.

The online version of this article includes the following source data for figure 5:

**Source data 1.** Evaluation of renal KIM-1 and TUNEL staining and albuminuria of mouse models.

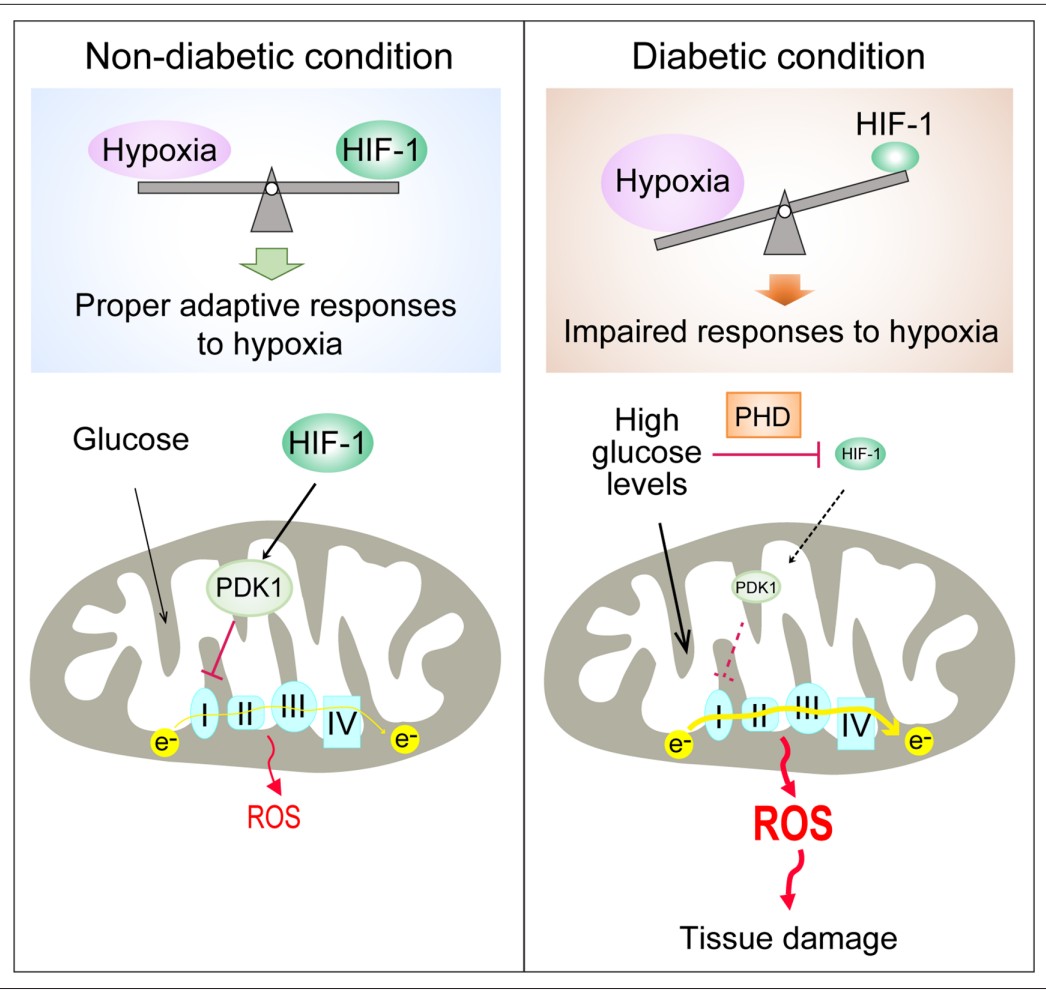

**Figure 6.** Repression of HIF-1 contributes to increased mitochondrial ROS production in diabetes. Under non-diabetic conditions (left panel), HIF-1 is induced by hypoxia and activates PDK1 expression which inhibits excess mitochondrial ROS production through inhibition of mitochondrial respiration. However, under diabetic conditions (right panel), HIF-1 is inhibited by high glucose levels through a PHD-dependent mechanism despite hypoxia. This results in decreased expression of PDK1, leading to increased mitochondrial respiration and excessive mitochondrial ROS production which causes tissue damage.

reports after exposure to cobalt, which also stabilizes HIF-1α (*Ohtomo et al., 2008*). The expression of the proximal tubular damage marker KIM-1, which in diabetic nephropathy becomes positive even before detection of albuminuria (*Nauta et al., 2011*; *Nordquist et al., 2015*), was not evident when ROS levels were suppressed by promoting HIF signaling in both animal models. This is in agreement with the absence of an increase of KIM-1 in diabetic kidneys where the renal oxygen levels were normalized (*Friederich-Persson et al., 2018*). Moreover, apoptosis, another classical marker of ROS damage in diabetic nephropathy (*Allen et al., 2003*), was reduced not only in DMOG-treated mIMCD3 cells exposed to high glucose concentrations in hypoxia but also in DMOG-treated *Lepr^{db/db}* mice and diabetic *Egln1^{+/-}* mice. Thus, promoting HIF-1 is a promising therapeutic strategy to prevent or treat even other chronic diabetes complications since excessive production of mitochondrial ROS is a key common driver of diabetic complications (*Charlton et al., 2020*; *Giacco and Brownlee, 2010*).

In conclusion, we demonstrate that the PHD-dependent HIF-1 repression induced by high glucose concentrations contributes to excessive production of mitochondrial ROS in diabetes, which is mediated by increased mitochondrial respiration secondary to the inhibition of HIF-1 target gene PDK1 (*Figure 6*). Promoting HIF-1 function is sufficient to normalize ROS levels during hyperglycemia and protects against diabetic nephropathy, making HIF-1 signaling an attractive therapeutic option for

diabetes complications. This is a timely finding, given that the first PHD inhibitor that can activate HIF-1 has been recently approved for clinical use (*Chen et al., 2019*).

## Acknowledgements

We thank to Valeria Alferova and Anette Landström from Karolinska Institutet, and Natasha Widen, Kajsa Sundqvist and Anette Härström from Karolinska University Hospital for excellent technical assistance.

## Additional information

### Funding

| Funder | Grant reference number | Author |
|---|---|---|
| Vetenskapsrådet | | Sergiu-Bogdan Catrina |
| Stockholms Läns Landsting | | Sergiu-Bogdan Catrina |
| Stockholm Regional Research Foundation | | Sergiu-Bogdan Catrina |
| Bert von Kantzows Foundation | | Sergiu-Bogdan Catrina |
| Swedish Society of Medicine | | Sergiu-Bogdan Catrina |
| Kung Gustaf V:s och Drottning Victorias Frimurarestifelse | | Sergiu-Bogdan Catrina |
| Karolinska Institute's Research Foundations | | Sergiu-Bogdan Catrina |
| Strategic Research Programme in Diabetes | | Sergiu-Bogdan Catrina |
| Erling-Persson Family Foundation | | Sergiu-Bogdan Catrina |
| Vetenskapsrådet | 2020-01645 | Tomas A Schiffer |
| Swedish Heart and Lung Foundation | 20210431 | Tomas A Schiffer |
| ERC consolidator grant | 773208 | Massimiliano Mazzone |

The funders had no role in study design, data collection and interpretation, or the decision to submit the work for publication.

### Author contributions

Xiaowei Zheng, Conceptualization, Data curation, Formal analysis, Funding acquisition, Investigation, Methodology, Project administration, Supervision, Validation, Visualization, Writing – original draft, Writing – review and editing; Sampath Narayanan, Cheng Xu, Data curation, Formal analysis, Investigation, Methodology, Validation, Visualization, Writing – original draft, Writing – review and editing; Sofie Eliasson Angelstig, Allan Zhao, Data curation, Formal analysis, Investigation, Validation, Writing – review and editing; Jacob Grünler, Data curation, Investigation, Methodology, Validation, Writing – review and editing; Alessandro Di Toro, Methodology, Writing – review and editing; Luciano Bernardi, Conceptualization, Methodology, Resources, Writing – review and editing; Massimiliano Mazzone, Peter Carmeliet, Resources, Writing – review and editing; Marianna Del Sole, Elisabete A Forsberg, Neda Rajamand Ekberg, Investigation, Writing – review and editing; Giancarlo Solaini, Writing – review and editing; Ao Zhang, Data curation, Investigation, Writing – review and editing; Kerstin Brismar, Conceptualization, Funding acquisition, Resources, Supervision, Writing – review and editing; Tomas A Schiffer, Data curation, Investigation, Methodology, Resources, Writing – review and editing; Ileana Ruxandra Botusan, Investigation, Writing – original draft, Writing – review and editing;

Fredrik Palm, Conceptualization, Funding acquisition, Resources, Supervision, Visualization, Writing – original draft, Writing – review and editing; Sergiu-Bogdan Catrina, Conceptualization, Funding acquisition, Project administration, Resources, Supervision, Visualization, Writing – original draft, Writing – review and editing

## Author ORCIDs
Xiaowei Zheng http://orcid.org/0000-0002-2648-1119
Allan Zhao http://orcid.org/0000-0002-2492-0923
Alessandro Di Toro http://orcid.org/0000-0001-7625-1103
Peter Carmeliet http://orcid.org/0000-0001-7961-1821
Neda Rajamand Ekberg http://orcid.org/0000-0001-5597-2593
Sergiu-Bogdan Catrina http://orcid.org/0000-0002-6914-3902

## Ethics
The clinical study was approved by the Regional Ethical Review Board in Stockholm, Sweden, and carried out in accordance with the principles of the Declaration of Helsinki. All participants in the study provided informed consent.

The experimental animal procedure was approved by the North Stockholm Ethical Committee for the Care and Use of Laboratory Animals (ethical permission N250/15, N60/15, and N179/16).

## Decision letter and Author response
Decision letter https://doi.org/10.7554/eLife.70714.sa1
Author response https://doi.org/10.7554/eLife.70714.sa2

---

# Additional files

## Supplementary files
- Transparent reporting form
- Source data 1. Unedited blots.

## Data availability
All data generated or analysed during this study are included in the manuscript and supporting files. Source data files have been provided for all the figures and tables.

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

# Appendix 1

## Appendix 1—key resources table

| Reagent type (species) or resource | Designation | Source or reference | Identifiers | Additional information |
|---|---|---|---|---|
| Strain, strain background (*Mus musculus*; male) | BKS(D)-Lepr<sup>db</sup>/JOrlRj, *Lepr<sup>db/db</sup>* diabetic mice | Janvier Labs | RRID:MGI:6293869 | |
| Strain, strain background (*Mus musculus*; male) | C57BL/6JRj Mouse | Janvier Labs | RRID:MGI:5751862 | |
| Strain, strain background (*Mus musculus*; male) | *Egln1<sup>+/-</sup>* and wild-type mice | Own colony | PMID:19217150 **Mazzone et al., 2009** | |
| Cell line (*Mus musculus*) | mIMCD-3 cell line | ATCC | Cat#: CRL-2123; RRID:CVCL_0429 | |
| Transfected construct (*M. musculus*) | siRNA to mouse *VHL* | Qiagen | Gene Solution siRNA (Cat#: 1027416) | Target sequence: TCCGAGATTGATCTACACATA |
| Transfected construct (*M. musculus*) | AllStars Negative Control siRNA | Qiagen | Cat#: 1027280 | |
| Antibody | anti-HIF-1alpha (Rabbit polyclonal) | GeneTex | Cat#: GTX127309; RRID:AB_2616089 | ICC(1:200) IHC(1:100) |
| Antibody | anti-KIM-1 (Rabbit polyclonal) | Novus Biologicals | Cat#: NBP1-76701; RRID:AB_11037459 | IHC(1:50) WB(1:500) |
| Antibody | Goat anti-Rabbit Secondary Antibody, Alexa Fluor 594 | Thermo Fisher Scientific | Cat#: A-11037; RRID:AB_2534095 | ICC (1:500) IHC (1:500) |
| Antibody | Goat anti-Rabbit Secondary Antibody, Alexa Fluor 488 | Thermo Fisher Scientific | Cat#: A-11008; RRID:AB_143165 | IHC (1:500) WB (1:500) |
| Antibody | anti-HIF-1alpha (Rabbit polyclonal) | Novus Biologicals | Cat#: NB100-479; RRID:AB_10000633 | WB: 1:500 |
| Antibody | anti-Histone H3 (Rabbit polyclonal) | Abcam | Cat#: ab1791; RRID:AB_302613 | WB: 1:5,000 |
| Antibody | anti-α-tubulin (mouse monoclonal) | Abnova | Cat#: MAB11106; RRID:AB_2888691 | WB:1:1,000 |
| Antibody | IRDye 800 goat anti-rabbit Secondary Antibody | LI_COR Biosciences | Cat#: 925–32211; RRID:AB_2651127 | WB:1:20,000 |
| Antibody | IRDye 680 goat anti-mouse Secondary Antibody | LI_COR Biosciences | Cat#: 925–68070; RRID:AB_2651128 | WB:1:20,000 |
| Recombinant DNA reagent | pCMV3-FLAG-PDK1 | Sino Biological Inc | Cat#: HG12312-NF | Plasmid encoding FLAG-tagged human PDK1 |
| Recombinant DNA reagent | pCMV3-GFP-FLAG-PDK1 | This paper | | Plasmid encoding GFP-fused FLAG-tagged human PDK1 |
| Sequence-based reagent | Mouse *PDK1*_F | This paper | PCR primers | AGTCCGTTGTCCTTATGAG |
| Sequence-based reagent | Mouse *PDK1*_R | This paper | PCR primers | CAGAACATCCTTGCCCAG |
| Sequence-based reagent | Mouse *BNIP3*_F | This paper | PCR primers | AACAGCACTCTGTCTGAGG |
| Sequence-based reagent | Mouse *BNIP3*_R | This paper | PCR primers | CCGACTTGACCAATCCCA |
| Sequence-based reagent | Mouse *PGK1*_F | This paper | PCR primers | AGTCCGTTGTCCTTATGAG |

*Appendix 1 Continued on next page*

*Appendix 1 Continued*

| Reagent type (species) or resource | Designation | Source or reference | Identifiers | Additional information |
|---|---|---|---|---|
| Sequence-based reagent | Mouse *PGK1_R* | This paper | PCR primers | CAGAACATCCTTGCCCAG |
| Sequence-based reagent | Mouse *SDF-1alpha_F* | This paper | PCR primers | GAGAGCCACATCGCCAGAG |
| Sequence-based reagent | Mouse *SDF-1alpha_R* | This paper | PCR primers | TTTCGGGTCAATGCACACTTG |
| Sequence-based reagent | Mouse *Egln1_F* | This paper | PCR primers | GGGCAACTACAGGATAAACGG |
| Sequence-based reagent | Mouse *Egln1_R* | This paper | PCR primers | CTCCACTTACCTTGGCGT |
| Sequence-based reagent | Mouse *GLUT3_F* | This paper | PCR primers | TCATCTCCATTGTCCTCCAG |
| Sequence-based reagent | Mouse *GLUT3_R* | This paper | PCR primers | CCAGGAACAGAGAAACTACAG |
| Sequence-based reagent | Mouse *ACTB_F* | This paper | PCR primers | AAGATCAAGATCATTGCTCCTC |
| Sequence-based reagent | Mouse *ACTB_R* | This paper | PCR primers | GGACTCATCGTACTCCTG |
| Sequence-based reagent | Mouse *HMBS_F* | This paper | PCR primers | CCTGTTCAGCAAGAAGATGGTC |
| Sequence-based reagent | Mouse *HMBS_R* | This paper | PCR primers | AGAAGTAGGCAGTGGAGTGG |
| Sequence-based reagent | Mouse *VHL_F* | This paper | PCR primers | CATCACATTGCCAGTGTATACCC |
| Sequence-based reagent | Mouse *VHL_R* | This paper | PCR primers | GCTGTATGTCCTTCCGCAC |
| Commercial assay or kit | MycoAlert PLUS mycoplasma detection kit | LONZA | Cat#: LT07-218 | |
| Commercial assay or kit | Dual-Luciferase Reporter Assay System | Promega | Cat#: E1960 | |
| Commercial assay or kit | Annexin V-FITC / 7-AAD kit | Beckman Coulter | Cat#: IM3614 | |
| Commercial assay or kit | Caspase-Glo 3/7 assay kit | Promega | Cat#: G8091 | |
| Commercial assay or kit | Quant-iT dsDNA High-Sensitivity Assay Kit | Thermo Fisher Scientific | Cat#: Q33120 | |
| Commercial assay or kit | Lipofectamine RNAiMAX Transfection Reagent | Thermo Fisher Scientific | Cat#: 13778075 | |
| Commercial assay or kit | MitoSOX Red Mitochondrial Superoxide Indicator, for live-cell imaging | Thermo Fisher Scientific | Cat#: M36008 | |
| Commercial assay or kit | ProLong Gold Antifade Mountant with DAPI | Thermo Fisher Scientific | Cat#: P36935 | |
| Commercial assay or kit | DAPI | Thermo Fisher Scientific | Cat#: D1306 | |
| Commercial assay or kit | Hypoxyprobe–1 Omni Kit | Hypoxyprobe, Inc | Cat#: HP1-XXX | |
| Commercial assay or kit | Tyramide Superboost kit | Thermo Fisher Scientific | Cat#: B40943 | |
| Commercial assay or kit | OxiSelectTM HNE Adduct Competitive ELISA kit | Cell Biolabs | STA838 | |
| Commercial assay or kit | DC Protein Assay | BIO-RAD | Cat#: 5000111 | |
| Commercial assay or kit | miRNeasy Mini kit | Qiagen | Cat#: 217,004 | |

*Appendix 1 Continued*

| Reagent type (species) or resource | Designation | Source or reference | Identifiers | Additional information |
|---|---|---|---|---|
| Commercial assay or kit | High-Capacity cDNA Reverse Transcription Kit | Thermo Fisher Scientific | Cat#: 4368814 | |
| Commercial assay or kit | SYBR Green Master Mix | Thermo Fisher Scientific | Cat#: 4367659 | |
| Commercial assay or kit | Bradford Protein Assay | BIO-RAD | Cat#: 5000001 | |
| Commercial assay or kit | In Situ Cell Death Detection Kit | Roche | Cat#: 11684817910 RRID:AB_2861314 | |
| Commercial assay or kit | DCA Microalbumin/ Creatinine Urine Test | Siemens Healthcare GmbH | Cat#: 01443699 | |
| Chemical compound, drug | CPH (1-hydroxy-3-carboxy-pyrrolidine) | Noxygen Science Transfer & Diagnostics GmbH | Cat#: NOX-01.1–50 mg | |
| Chemical compound, drug | EPR-grade Krebs HEPES buffer | Noxygen Science Transfer & Diagnostics GmbH | Cat#: NOX-7.6.1–500 ml | |
| Chemical compound, drug | Deferoxamine | Noxygen Science Transfer & Diagnostics GmbH | Cat#: NOX-09.1–100 mg | |
| Chemical compound, drug | DETC (diethyldithiocarbamate) | Noxygen Science Transfer & Diagnostics GmbH | Cat#: NOX-10.1–1 g | |
| Chemical compound, drug | DMOG (Dimethyloxalylglycine) | Frontier Specialty Chemicals | Cat#: D1070 | |
| Chemical compound, drug | cOmplete, Mini, EDTA-free Protease Inhibitor Cocktail | Roche | Cat#: 11836170001 | |
| Chemical compound, drug | Formaldehyde solution | Sigma | Cat#: F8775 | |
| Chemical compound, drug | Streptozotocin | Sigma | Cat#: S0130 | |
| Chemical compound, drug | Sudan Black B | Sigma | Cat#: 199,664 | |
| Software, algorithm | FlowJo | FlowJo | RRID:SCR_008520 | |
| Software, algorithm | Image-Pro Premier v9.2 | Media Cybernetics | | |
| Software, algorithm | ImageJ | ImageJ | RRID:SCR_003070 | |
| Software, algorithm | GraphPad Prism | GraphPad Prism | RRID:SCR_002798 | |
| Other | Dulbecco's Modified Eagle's Medium | Thermo Fisher Scientific | 31885–023 | |

