## [Editor Report]

The paper is novel, informative, and with interesting translational implications. This paper will be of interest to scientists interested in diabetes and its complications, as well as the wider field of hypoxia biology. It provides evidence to understand why diabetes causes damage to multiple tissues when oxygen supply becomes limited.

---

## [Decision Letter]

**Decision letter after peer review:**

Thank you for submitting your article "Repression of Hypoxia-Inducible Factor-1 Contributes to Increased Mitochondrial Reactive Oxygen Species Production in Diabetes" for consideration by *eLife*. Your article has been reviewed by 3 peer reviewers, including Ernestina Schipani as Reviewing Editor and Reviewer #1, and the evaluation has been overseen by Mone Zaidi as the Senior Editor.

Essential revisions:

The reviewers have listed numerous strengths. Weaknesses were also identified. Please, address concerns and comments point-by-point.

Of note, additional experiments are less important than modifications to current data and text. Reanalysis and modifying text are essential to support the claims of the paper.

*Reviewer #1:*

In this study, Zheng and colleagues report the novel findings that in diabetic models hyperglycemia suppress HIF1a in a PHD-dependent manner, and this in turn leads to increased mitochondrial ROS and cell death. Both the increased ROS and the cell death are prevented by increasing HIF1a activity either pharmacologically or genetically.

The paper is novel, informative, and with interesting translational implications. The authors used a variety of in vitro and in vivo models for the testing of their hypothesis, with special emphasis on a model of diabetic nephropathy.

However, a few issues need to be addressed in order to strengthen the authors' conclusions and their biological significance.

1. Suppressing HIF1a should also increase mitochondrial oxygen consumption and, thus, intracellular hypoxia. Along those lines, it would be helpful if the authors could show the pimonidazole data per se, in addition to the ratio HIF1a/pimonidazole. Increasing intracellular hypoxia could be an alternative mechanism that may promote cell death upon HIF1a suppression.

2. The authors do not provide unequivocal evidence that the increased mitochondrial ROS are responsible for the cell death upon exposure to hyperglycemia. Along those lines, use of reducing agents such as NAC could be helpful and informative.

3. Is the cell death documented by the authors caspase-3 dependent?

4. All western blot data should be properly quantified.

5. In the various genetic models, inactivation or increased expression of the gene of interest should be properly documented.

*Reviewer #2:*

The manuscript by Zheng et al. addresses the question of whether abnormal HIF stabilisation in response to hypoxia could be responsible for the increased ROS generation in diabetes, and whether hyperglycaemia could be the driver for the ROS damage and cell death in hypoxia. Diabetes complications across multiple organs are associated with oxidative damage and are accompanied by restricted oxygen delivery due to vascular dysfunction. Thus, understanding how these two phenotypes are linked by diabetes is an important question for advancing our understanding of diabetes complications and to identify novel therapeutic targets.

Strengths

This study uses multiple different techniques to address this question, using cell culture, animal models as well as human samples.

Mechanistic conclusions are based on both genetic and pharmacological approaches, adding strength to the key findings.

This is an excellent continuation of high-quality work conducted by this group in this field. The conclusions have far reaching consequences within the field.

Weaknesses

Although the authors establish a translational pipeline for their findings, from cells to animals to humans, the findings from the human studies could be interpreted in alternative ways to that presented. Cells cultured for 24hrs in hypoxia have sufficient time to activate the HIF transcription factor, its downstream targets and result in a functional outcome for the cell. Humans exposed to hypoxia for 1 hr will not have the same time to induce the HIF-dependent effects, and other mechanisms will be at play. To some extent this is due to the difficulties of performing mechanistic studies in humans, however, care should be taken in not oversimplifying findings and instead highlighting alternative conclusions.

Reactive oxygen species are a group of molecules comprising different species with different reactivities. It is important to be clear which aspects of ROS and their downstream damage are being measured with these methodologies.

Care must be taken to ensure the correct statistical analysis is performed based on the groups assessed, the study design and the scientific question being asked.

My major concern relates to the conclusions drawn from the human studies and their direct relevance to the cell/animal work.

1. Results – human study. This is a really interesting and important finding that the diabetic patients have greater ROS generation in response to hypoxia. However, this is an acute hypoxic response and will not be related to any HIF-related changes in mitochondrial function that form the core of the rest of the paper. These are two independent mechanisms/phenomena. I think this has to be addressed at some point in the paper, that the findings in patients cannot be explained by the mechanism presented in the rest of the manuscript, and are more likely due to other acute hypoxic factors. For example, in the discussion the authors state 'An optimal HIF-1 response during hypoxia, as seen in the control subjects in our study" – this is false – 1 hr is not sufficient to induce a HIF response that has a functional outcome that could have been measured in the blood of the patients.

2. Abstract – nonspecific terminology needs correcting to make the abstract specific to the findings. Terms such as "the dynamics of ROS levels" and "was followed by functional consequences" don't really tell the reader what was found in the paper. Also, please don't use the term "findings are highly significant" as this could relate to statistics.

3. Introduction – paragraph 1 – The final sentence about ROS generation being related to increased glucose availability is not relevant to all tissues – in heart for example there is decreased intracellular glucose availability. Please correct to make this broad statement more tissue specific.

4. Methods – Evaluation of ROS levels in kidney. 4HNE does not detect ROS – it measures oxidative damage as evident by lipid peroxidation. It is incorrect in the methods and results to describe this as ROS levels.

5. Methods – Clinical study – please describe how the patients were given 5 hypoxic episodes – methodological detail is lacking.

6. Statistics – I am confused about some of the choices for statistical analysis. In figure 1 why was an unpaired t-test used when the same patient has been tested at 0 and 4 hrs, this should be a paired t-test. Figure 2 – I'm unclear why a repeated measures ANOVA was used, when these were independent groups of cells, not the same cells measured at different points/conditions (this shouldn't be repeated measures). Figures 4 and 5 – when there are groups with 2 factors varying – genotype (wt vs PHD) and diabetic status (Ctl and db) – this should be a 2-way ANOVA not a one-way ANOVA.

7. Figure 3 – panels B and C – could the authors show HIF blots for these same groups, to compliment the mitosox intensity, so its possible to relate ROS to the degree of HIF stabilisation in these genetic cell models.

8. Figure 4 – could the authors please comment on why respiration in the PHD2 mutant isn't below the WT? One would expect the HIF stabilisation in this model to decrease oxygen consumption.

9. Discussion – 3rd paragraph. The authors have made incorrect conclusions regarding their respiration data. They state that "our results indicate that the role of HIF on ROS is limited to complex I". What the data shows is that when respiring through the ETC that involves all of complexes I, II, III, and IV – with substrate feeding in at the start and oxygen consumed at the end – that there is a HIF dependent effect. They cannot conclude it is at complex I as they haven't independently assessed complex II (independent of complex I), complex III and separately complex IV (which would need to be done with succinate alone, DHQ and TMPD as 3 separate experiments). The data shows the HIF effect resides at some point between Complex I and Complex IV.

10. Table 3 – blood glucose before STZ – is this significantly different?

11. Page 9 – PDK1 only inhibits flux through the TCA is exclusively metabolising glucose – not the case if metabolising amino acids or fatty acids.

*Reviewer #3:*

Strengths:

1. Oxidative stress and damage is central to pathology of diabetic complications. As mitochondria are a key generator of ROS-mediated damage, the authors nicely connect glucose-dependent HIF1a deficiency with the development of mitochondrial ROS-mediated damage.

2. Approaches to understand and target oxidant stress in diabetic complications are elusive and the authors nicely delineate a PHD2-dependent mechanism by which HIF1a levels decline, subsequently giving rise to renal dysfunction.

3. Notably, improvement of HIF1a levels by PHD2 pharmacologic or genetic inhibition appears to ameliorate renal injury independent of glycemic control. This could be of high translational value as therapies to ameliorate diabetic nephropathy do not exist currently in the clinic.

Weaknesses:

1. The mechanism by which hyperglycemia precipitates PHD2-dependent HIF1a degradation and activation of renal injury is not clear.

2. The induction of mitochondrial damage to elicit mitochondrial ROS and subsequent renal compromise downstream of HIF1a deficiency is proposed to be via PDK1. It would strengthen the paper if the PDK1-dependent mechanism was further solidified.

3. Clinical data on patients with uncontrolled T1D and impaired circulating hypoxia responses are very interesting, but it is unclear how they directly relate to the renal specific findings presented in the remainder of the manuscript.

The study presented by Zheng and colleagues is well written and serves to highlight the important role of ROS in the development of diabetic kidney disease. The studies are interesting, well written, and nicely performed. Further, the translational implications of PHD2 inhibition to potentially ameliorate diabetic nephropathy are highly appealing. However, a central concern is that some of the major observations of this study, while interesting, appear to be consistent with other previous observations in the field. Indeed, several groups have observed a depletion of HIF1a leads to renal demise. Further, PHD2 genetic and pharmacologic inhibition have been shown by several groups to protect against renal injury. Finally, the importance of mitochondrial ROS in diabetic kidney disease is also well known. While this study nicely weaves the concept together, a concern is the results are mostly confirmatory in nature.

Thus, several approaches could be employed to enhance the novelty of the observations within the manuscript.

1. The authors note that glucose reduces HIF1a levels through a PHD2 dependent mechanism, yet in Figure 4 supplement 2, PHD2 levels do not rise in db/db mice. It would substantially improve the manuscript if the mechanisms underlying increased PHD2 activity/HIF1a degradation were reconciled. Is enzymatic activity enhanced? Is it possible that VHL targets HIF1a for degradation more rapidly? Garcia-Pastor et al. (Sci Reports 2019) previously published that glucose inhibits the interaction of HIF1a with Hsp90 in cell lines. Does this occur in primary cells as well?

2. The authors implicate loss of PDK1 activity downstream of HIF1a deficiency as a key mechanism underlying the increase in complex I activity and subsequent mitochondrial ROS. However, reductions in PDK1 gene expression are modest at best. This mechanism, while based upon a highly interesting study from Kim and colleagues in Cell Metabolism, is not well substantiated in the author's study. More conclusive experimentation reconciling this pathway's importance is crucial for the development of the mechanism underlying the authors' model.

3. Increases in mitochondrial ROS in diabetic nephropathy have also been thought to be related to effects on the Nrf2 anti-oxidant response, which is not discussed in this study. It would be helpful to address the potential roles of Nrf2 in the pro-oxidant responses surveyed in their mouse and cell based models.

4. Clinical data on T1D patients, while interesting, is of unclear utility in this manuscript. Do these patients have renal disease? While HbA1c levels are higher in T1D patients versus non-T1D patients, are ROS responses improved in T1D patients with normal glycemic control? Further, as ROS levels are often transient in the blood stream, what are the ambient blood glucose levels during the period of the EPR assays?

---

## [Author Response]

Reviewer #1:In this study, Zheng and colleagues report the novel findings that in diabetic models hyperglycemia suppress HIF1a in a PHD-dependent manner, and this in turn leads to increased mitochondrial ROS and cell death. Both the increased ROS and the cell death are prevented by increasing HIF1a activity either pharmacologically or genetically.The paper is novel, informative, and with interesting translational implications. The authors used a variety of in vitro and in vivo models for the testing of their hypothesis, with special emphasis on a model of diabetic nephropathy.However, a few issues need to be addressed in order to strengthen the authors' conclusions and their biological significance.1. Suppressing HIF1a should also increase mitochondrial oxygen consumption and, thus, intracellular hypoxia. Along those lines, it would be helpful if the authors could show the pimonidazole data per se, in addition to the ratio HIF1a/pimonidazole. Increasing intracellular hypoxia could be an alternative mechanism that may promote cell death upon HIF1a suppression.

The levels of hypoxia evaluated by pimonidazole staining were added as Figure 4-supplement figure 1. It is true that the levels of intracellular hypoxia were increased in diabetic kidneys compared with the animals without diabetes that confirms previous observations (Rosenberger, Khamaisi et al. 2008, PMID: 17914354) and were improved by HIF induction in both animal models. Their potential contribution to tissue injury was added in Discussion (the first paragraph of page 26).

2. The authors do not provide unequivocal evidence that the increased mitochondrial ROS are responsible for the cell death upon exposure to hyperglycemia. Along those lines, use of reducing agents such as NAC could be helpful and informative.

We thank to the reviewer for the constructive suggestion. Since we have found that the cell death induced by high glucose levels in hypoxia is dependent on caspase-3/7 (next question), we have verified the effect of NAC on the caspase-3/7 activity in mIMCD3 cells. As shown in Figure 2G, the caspase-3/7 activity increased in mIMCD3 cells exposed to high glucose concentrations and hypoxia, and was inhibited by NAC pre-treatment. These results indicate that the increased mitochondrial ROS is responsible for cell apoptosis upon exposure to hyperglycemia and hypoxia. This result has been added to the manuscript as Figure 2G.

3. Is the cell death documented by the authors caspase-3 dependent?

The caspase-3/7 activity has been evaluated in mIMCD3 cells exposed to normal (5.5 mM) or high (30 mM) glucose levels in normoxia (N) or hypoxia (H) using Caspase-Glo 3/7 Assay kit from Promega. As shown in Figure 2F, high glucose levels in hypoxia enhanced caspase 3/7 activity in mIMCD3 cells, which could be inhibited by DMOG treatment. These results suggest that the apoptosis induced by high glucose levels in hypoxia is dependent on caspase-3 and -7 and are added to the manuscript as Figure 2F.

4. All western blot data should be properly quantified.

All the western blot data were quantified as suggested. The quantifications have been added below each western blot.

5. In the various genetic models, inactivation or increased expression of the gene of interest should be properly documented.

We thank the reviewer for very good suggestion. The expression of VHL gene in cells transfected with control siRNA and VHL siRNA is shown in Figure 3B, and the endogenous nuclear HIF-1α protein expression was analysed using fluorescent immunocytochemistry (Figure 3C). Expression of GFP and GFP-HIF-1α were verified using confocal microscopy and the nuclear HIF-1α expression in GFP-HIF-1α-expressing cells was further confirmed by immunocytochemistry using anti-HIF-1α antibody (Figure 3E). Egln1 gene expression in healthy and diabetic wild-type and *Egln1*^+/-^ mice is shown in Figure 4 —figure supplement 3B.

Reviewer #2:[…]My major concern relates to the conclusions drawn from the human studies and their direct relevance to the cell/animal work.1. Results – human study. This is a really interesting and important finding that the diabetic patients have greater ROS generation in response to hypoxia. However, this is an acute hypoxic response and will not be related to any HIF-related changes in mitochondrial function that form the core of the rest of the paper. These are two independent mechanisms/phenomena. I think this has to be addressed at some point in the paper, that the findings in patients cannot be explained by the mechanism presented in the rest of the manuscript, and are more likely due to other acute hypoxic factors. For example, in the discussion the authors state 'An optimal HIF-1 response during hypoxia, as seen in the control subjects in our study" – this is false – 1 hr is not sufficient to induce a HIF response that has a functional outcome that could have been measured in the blood of the patients.

We agree with the reviewer that exposure for 1 hour to hypoxia is not sufficient for the functional outcome of HIF-1-related genes. We believe however that this short exposure to hypoxia unmasks the impaired function of HIF-1 that is already present in the subjects with diabetes before exposure to hypoxia. Our hypothesis is sustained by the findings that the plasma levels of microRNA-210 (which is a target uniquely regulated by HIF-1) are lower in the subjects with diabetes (26.7 ± 14.5 %, *P*<0.05) than in controls (100 ± 52.5 %) before exposure to hypoxia. However, we cannot exclude other sources of ROS independent of HIF-1 that are activated by short exposure to hypoxia, either from mitochondria (Waypa, Marks et al. 2013. PMID: 23328522) (Hernansanz-Agustin, Choya-Foces et al. 2020. PMID: 32728214), or elsewhere i.e. NADPH oxidases (Weissman, Tadic et al. 2000. PMID: 11000128). This has been added to the Discussion in the manuscript (2^nd^ paragraph of page 24).

2. Abstract – nonspecific terminology needs correcting to make the abstract specific to the findings. Terms such as "the dynamics of ROS levels" and "was followed by functional consequences" don't really tell the reader what was found in the paper. Also, please don't use the term "findings are highly significant" as this could relate to statistics.

We made the changes suggested by the reviewer.

3. Introduction – paragraph 1 – The final sentence about ROS generation being related to increased glucose availability is not relevant to all tissues – in heart for example there is decreased intracellular glucose availability. Please correct to make this broad statement more tissue specific.

The sentence was changed as suggested.

4. Methods – Evaluation of ROS levels in kidney. 4HNE does not detect ROS – it measures oxidative damage as evident by lipid peroxidation. It is incorrect in the methods and results to describe this as ROS levels.

We completely agree with the reviewer and changed the description accordingly.

5. Methods – Clinical study – please describe how the patients were given 5 hypoxic episodes – methodological detail is lacking.

The details of the hypoxia exposures were added (page 7).

6. Statistics – I am confused about some of the choices for statistical analysis. In figure 1 why was an unpaired t-test used when the same patient has been tested at 0 and 4 hrs, this should be a paired t-test. Figure 2 – I'm unclear why a repeated measures ANOVA was used, when these were independent groups of cells, not the same cells measured at different points/conditions (this shouldn't be repeated measures). Figures 4 and 5 – when there are groups with 2 factors varying – genotype (wt vs PHD) and diabetic status (Ctl and db) – this should be a 2-way ANOVA not a one-way ANOVA.

Unpaired t-test was applied in Figure 1 because some samples were missing that precludes the use of paired t-test. We agree that ordinary one-way ANOVA test is appropriate to be used for Figure 2 and changed in consequence. We agree that two-way ANOVA test should be applied in the experiments that have two independent variables (genotype and diabetic status) and we changed accordingly.

7. Figure 3 – panels B and C – could the authors show HIF blots for these same groups, to compliment the mitosox intensity, so its possible to relate ROS to the degree of HIF stabilisation in these genetic cell models.

We agree with the reviewer that it is important to show the HIF-1α expression levels in these experiments. We have therefore documented the nuclear HIF-1α expression in the conditions where ROS levels were decreased. These results have been added to Figure 3.

8. Figure 4 – could the authors please comment on why respiration in the PHD2 mutant isn't below the WT? One would expect the HIF stabilisation in this model to decrease oxygen consumption.

Very interesting question! We believe that non-diabetic condition is a non-challenged state, where the local oxygen levels in kidney are similar between WT and *Egln1*^+/-^ mice (PHD mutant), as shown by similar pimonidazole staining levels (Figure 4 —figure supplement 1B). Under this condition, HIF-1 signaling is under proper regulation as illustrated by similar HIF-1α (Figure 4C) and HIF-1 target gene e.g. GLUT3 (Figure 4 —figure supplement 3) expression levels, therefore oxygen consumption rate is comparable.

The effects of HIF-1α stabilisation secondary to the genetic manipulation of *Egln1* (encoding PHD2) is however uncovered in diabetes, where it maintains an adequate HIF-1 level in opposition to WT mice with diabetes and in consequence to have a lower respiration.

9. Discussion – 3rd paragraph. The authors have made incorrect conclusions regarding their respiration data. They state that "our results indicate that the role of HIF on ROS is limited to complex I". What the data shows is that when respiring through the ETC that involves all of complexes I, II, III, and IV – with substrate feeding in at the start and oxygen consumed at the end – that there is a HIF dependent effect. They cannot conclude it is at complex I as they haven't independently assessed complex II (independent of complex I), complex III and separately complex IV (which would need to be done with succinate alone, DHQ and TMPD as 3 separate experiments). The data shows the HIF effect resides at some point between Complex I and Complex IV.

We thank to the reviewer for the comments. Indeed, the experimental design used does not allow to rule out other mitochondrial ROS generating contributors and we changed the statement in consequence.

10. Table 3 – blood glucose before STZ – is this significantly different?

The blood glucose levels before administration of STZ are not different between wild type mice and *Egln1*^+/-^ mice using two-way ANOVA test.

11. Page 9 – PDK1 only inhibits flux through the TCA is exclusively metabolising glucose – not the case if metabolising amino acids or fatty acids.

We completely agree with the reviewer and we have made the appropriate correction.

Reviewer #3:[…]The study presented by Zheng and colleagues is well written and serves to highlight the important role of ROS in the development of diabetic kidney disease. The studies are interesting, well written, and nicely performed. Further, the translational implications of PHD2 inhibition to potentially ameliorate diabetic nephropathy are highly appealing. However, a central concern is that some of the major observations of this study, while interesting, appear to be consistent with other previous observations in the field. Indeed, several groups have observed a depletion of HIF1a leads to renal demise. Further, PHD2 genetic and pharmacologic inhibition have been shown by several groups to protect against renal injury. Finally, the importance of mitochondrial ROS in diabetic kidney disease is also well known. While this study nicely weaves the concept together, a concern is the results are mostly confirmatory in nature.Thus, several approaches could be employed to enhance the novelty of the observations within the manuscript.1. The authors note that glucose reduces HIF1a levels through a PHD2 dependent mechanism, yet in Figure 4 supplement 2, PHD2 levels do not rise in db/db mice. It would substantially improve the manuscript if the mechanisms underlying increased PHD2 activity/HIF1a degradation were reconciled. Is enzymatic activity enhanced? Is it possible that VHL targets HIF1a for degradation more rapidly? Garcia-Pastor et al. (Sci Reports 2019) previously published that glucose inhibits the interaction of HIF1a with Hsp90 in cell lines. Does this occur in primary cells as well?

We thank the reviewer for raising this important question. The mechanisms by which high glucose levels impair HIF-1 expression and function are still incompletely unraveled. We agree with the reviewer that, even though we bring pharmacological and genetic evidence about the importance of PHD-dependent degradation for the HIF-1 repression in diabetes, we did not dissect the specific mechanisms facilitating the PHD2-mediated regulation of HIF-1 in diabetes, a subject that warrants further investigation. Our results have shown that silencing VHL gene can stabilize HIF-1α and reduce mitochondrial ROS levels in cells exposed to high glucose levels and hypoxia (Figure 3B-3D), indicating that VHL contributes to the degradation of HIF-1α in diabetic conditions. We would like to clarify that we have not claimed that the inhibition of HIF-1 in diabetes is specific to the PHD2-dependent HIF-1α degradation. The contribution of other mechanisms to the inhibition of HIF-1 signaling in diabetes needs to be further elucidated.

2. The authors implicate loss of PDK1 activity downstream of HIF1a deficiency as a key mechanism underlying the increase in complex I activity and subsequent mitochondrial ROS. However, reductions in PDK1 gene expression are modest at best. This mechanism, while based upon a highly interesting study from Kim and colleagues in Cell Metabolism, is not well substantiated in the author's study. More conclusive experimentation reconciling this pathway's importance is crucial for the development of the mechanism underlying the authors' model.

We thank the reviewer for the constructive comments. In order to further investigate the role of PDK1, we transfected plasmids encoding GFP or GFP-fused PDK1 (GFP-PDK1) in mIMCD3 cells exposed to high glucose levels and hypoxia (Figure 4I).The mitochondrial ROS levels were assessed by flow cytometry analysis of mitosox intensity in GFP- or GFP-PDK1-positive cells. As shown in Figure 4J, mitochondrial ROS overproduction in cells exposed to high glucose levels and hypoxia was diminished by PDK1 overexpression, suggesting that the increased mitochondrial ROS is at least partially mediated by the inhibition of HIF-1 target gene PDK1 in diabetes. These results have been added to the manuscript (page 22).

3. Increases in mitochondrial ROS in diabetic nephropathy have also been thought to be related to effects on the Nrf2 anti-oxidant response, which is not discussed in this study. It would be helpful to address the potential roles of Nrf2 in the pro-oxidant responses surveyed in their mouse and cell based models.

We thank the reviewer for the suggestion. The potential role of Nrf2 for diabetes nephropathy has been included in the discussion (page 25).

4. Clinical data on T1D patients, while interesting, is of unclear utility in this manuscript. Do these patients have renal disease? While HbA1c levels are higher in T1D patients versus non-T1D patients, are ROS responses improved in T1D patients with normal glycemic control? Further, as ROS levels are often transient in the blood stream, what are the ambient blood glucose levels during the period of the EPR assays?

The patients with T1D were without any detectable complication, including renal disease. The clinical study provides the proof of concept that the subjects with uncontrolled diabetes responded with an increase in ROS levels after exposure to hypoxia, while as ROS levels did not increase in subjects without diabetes.

We agree with the reviewer that a separate investigation on whether improvement of the metabolic control can affect the ROS response to hypoxia warrants further investigation.

In the present study, plasma glucose levels before hypoxia exposure (0h) were significantly higher in patients with diabetes (12.79 ± 6.62 mM) than in control subjects (4.66 ± 0.90 mM), as expected, since only patients with poor glycemic control have been included. After one hour of hypoxia exposure (1h), plasma glucose increased in healthy subjects by 0.64 ± 0.91 mM (*P*<0.05). This is in line with previous findings (e.g. PMID: 28087818), and may represent a mechanism to protect tissues during hypoxia by increasing fuel supply. Interestingly, plasma glucose levels decreased in patients with diabetes by 3.05 ± 5.03 mM (*P*<0.05). This opposite response of blood glucose is part of the impaired adaptive responses to hypoxia present in patients with diabetes, that is subject of our ongoing investigation.